# Fair Max–Min Diversity Maximization in Streaming and Sliding-Window Models [note 1]

**DOI:** 10.3390/e25071066

**Published:** 2023-07-14

**Authors:** Yanhao Wang, Francesco Fabbri, Michael Mathioudakis, Jia Li

**Affiliations:** 1School of Data Science and Engineering, East China Normal University, Shanghai 200062, China; 2Spotify, 08000 Barcelona, Spain; 3Department of Computer Science, University of Helsinki, 00560 Helsinki, Finland

**Keywords:** diversity maximization, group fairness, streaming algorithm, sliding-window algorithm

## Abstract

Diversity maximization is a fundamental problem with broad applications in data summarization, web search, and recommender systems. Given a set *X* of *n* elements, the problem asks for a subset *S* of k≪n elements with maximum diversity, as quantified by the dissimilarities among the elements in *S*. In this paper, we study diversity maximization with fairness constraints in streaming and sliding-window models. Specifically, we focus on the max–min diversity maximization problem, which selects a subset *S* that maximizes the minimum distance (dissimilarity) between any pair of distinct elements within it. Assuming that the set *X* is partitioned into *m* disjoint groups by a specific sensitive attribute, e.g., sex or race, ensuring fairness requires that the selected subset *S* contains $k_{i}$ elements from each group i∈[m]. Although diversity maximization has been extensively studied, existing algorithms for fair max–min diversity maximization are inefficient for data streams. To address the problem, we first design efficient approximation algorithms for this problem in the (insert-only) streaming model, where data arrive one element at a time, and a solution should be computed based on the elements observed in one pass. Furthermore, we propose approximation algorithms for this problem in the sliding-window model, where only the latest *w* elements in the stream are considered for computation to capture the recency of the data. Experimental results on real-world and synthetic datasets show that our algorithms provide solutions of comparable quality to the state-of-the-art offline algorithms while running several orders of magnitude faster in the streaming and sliding-window settings.

## 1. Introduction

Data summarization is a common approach to tackling the challenges of a large volume of data in data-intensive applications. That is because, rather than performing high-complexity analyses on the whole dataset, it is often beneficial to perform them on a representative and significantly smaller summary of the dataset, thus reducing the processing costs in terms of both running time and space usage. Typical techniques for data summarization [1] include sampling, sketching, coresets, and diverse data selection.

In this paper, we focus on diversity-aware data summarization, which finds application in a wide range of real-world problems. For example, in database query processing [2,3], web search [4,5], and recommender systems [6], the output might be too large to be presented to the user in its entirety, even after filtering the results by relevance. One feasible solution, then, is to present the user with a small but diverse subset that is easy to process and representative of the complete results. As another example, when training machine learning models on massive data, feature and subset selection is a standard method to improve efficiency. As indicated by [7,8], selecting diverse features or subsets can lead to a better balance between efficiency and accuracy. A key technical problem in such cases is diversity maximization [9,10,11,12,13,14,15,16,17,18,19,20].

In more detail, for a given set *X* of elements in some metric space and a size constraint *k*, diversity maximization asks for a subset of *k* elements with maximum diversity. Formally, diversity is quantified by a function that captures how well a subset spans the range of elements in *X*, and is typically defined in terms of distances or dissimilarities among elements in the subset. Prior studies [3,4,6,12] have suggested many different objectives of this kind. Two of the most popular ones are max–sum dispersion, which aims to maximize the sum of the distances between all pairs of elements in the selected subset *S*, and max–min dispersion, which aims to maximize the minimum distance between any pair of distinct elements in *S*. Figure 1 illustrates a selection of the 10 most diverse points from a two-dimensional point set with each of the two objectives for diversity maximization. As shown in Figure 1, max–sum dispersion tends to select “outliers” and may include highly similar elements in the solution, making it unsuitable for applications requiring more uniform coverage of the span of data. Therefore, we focus on diversity maximization with the objective of max–min dispersion, referred to as max–min diversity maximization, in this paper.

In addition to diversity, fairness in data summarization is also attracting increasing attention [8,21,22,23,24,25,26,27]. Several studies reveal that the biases with respect to (w.r.t.) sensitive attributes, such as sex, race, or age, in underlying datasets can be retained in the summaries and could lead to unfairness in data-driven social computational systems such as education, recruitment, and banking [8,23,26]. One of the most common notions for fairness in data summarization is group fairness [8,21,22,23,27], which partitions the dataset into *m* disjoint groups based on a specific sensitive attribute and introduces a fairness constraint that limits the number of elements from group *i* in the data summary to $k_{i}$ for every group i∈[m] (see Figure 2 for an illustrative example). However, most existing methods for diversity maximization cannot easily be adapted to satisfy such fairness constraints. Moreover, a few methods that can deal with fairness constraints are specific to max–sum diversity maximization [9,11,13]. To the best of our knowledge, the methods in [17,20] are the only means of max–min diversity maximization with fairness constraints.

Furthermore, since many applications of diversity maximization are in the realm of massive data analysis, it is essential to design efficient algorithms for processing large-scale datasets. The (insert-only) streaming and sliding-window models are well-recognized frameworks for big data processing. In the streaming model, an algorithm is only permitted to process each element in the dataset sequentially in one pass, is allowed to take time and space that are sublinear to or even independent of the dataset size, and is required to provide solutions of comparable quality to those returned by the offline algorithms. In the sliding-window model, the computation is further restricted to the latest *w* elements in the stream, and an algorithm is required to find good solutions in sublinear time and space w.r.t. the window size. However, the only known algorithms [17,20] for fair max–min diversity maximization are designed for the offline setting and are very inefficient in the streaming and sliding-window models.


**Our Contributions:** In this paper, we propose novel streaming and sliding-window algorithms for the max–min diversity maximization problem with fairness constraints. Our main contributions are summarized as follows:
We formally define the problem of fair max–min diversity maximization (FDM) in metric spaces. Then, we describe the existing streaming and sliding-window algorithms for (unconstrained) max–min diversity maximization [14]. In particular, we improve the approximation ratio of the existing streaming algorithm from $\frac{1-\varepsilon }{5}$ to $\frac{1-\varepsilon }{2}$ for any parameter ε∈(0,1) by refining the analysis of [14].We propose two novel streaming algorithms for FDM. Our first algorithm, called SFDM1, is $\frac{1-\varepsilon }{4}$-approximate for FDM when there are two groups in the dataset. It takes $O(\frac{k\log\Delta }{\varepsilon })$ time per element in the streaming processing, where Δ is the ratio of the maximum and minimum distances between any pair of elements, spends $O(\frac{k^{2}\log\Delta }{\varepsilon })$ time for post-processing, and stores $O(\frac{k\log\Delta }{\varepsilon })$ elements in memory. Our second algorithm, called SFDM2, is $\frac{1-\varepsilon }{3m+2}$-approximate for FDM with an arbitrary number *m* of groups. SFDM2 also takes $O(\frac{k\log\Delta }{\varepsilon })$ time per element in the streaming processing but requires a longer $O\left(\frac{k^{2}m\log\Delta }{\varepsilon }\cdot \left(m+\log^{2}k\right)\right)$ time for post-processing and stores $O(\frac{km\log\Delta }{\varepsilon })$ elements in memory.We further extend our two streaming algorithms to the sliding-window model. The extended SWFDM1 and SWFDM2 algorithms achieve approximation factors of Θ(1) and $\Theta (m^{-1})$ for FDM with m=2 and an arbitrary *m* when any Θ(1)-approximation algorithm for unconstrained max–min diversity maximization is used for post-processing. Additionally, their time and space complexities increase by a factor of $O(\frac{\log\Delta }{\varepsilon })$ compared with SFDM1 and SFDM2, respectively.Finally, we evaluate the performance of our proposed algorithms against the state-of-the-art algorithms on several real-world and synthetic datasets. The results demonstrate that our algorithms provide solutions of comparable quality for FDM to those returned by the state-of-the-art algorithms while running several orders of magnitude faster in the streaming and sliding-window settings.


A preliminary version of this paper is published in [28]. In this extended version, we make the following novel contributions with respect to [28]: (1) We propose two novel algorithms for FDM in the sliding-window model along with the implementation of an existing algorithm for unconstrained max–min diversity maximization in the sliding-window model [14]. Moreover, we analyze the approximation factors and complexities of the two algorithms for fair sliding-window diversity maximization; (2) We conduct more comprehensive examinations of our streaming algorithms by implementing and comparing them with a new offline baseline called FairGreedyFlow [20], which achieves a better approximation factor than previous offline algorithms. The additional results further confirm the superior performance of our streaming algorithms; (3) We conduct new experiments for FDM in the sliding-window setting to evaluate the performance of our sliding-window algorithms compared with the existing offline algorithms. The new experimental results validate their efficiency, effectiveness, and scalability.
**Paper Organization:** The rest of this paper is organized as follows. The related work is reviewed in Section 2. In Section 3, we introduce the basic concepts and formally define the FDM problem. In Section 4, we first propose our streaming algorithms for FDM. In Section 5, we further design our sliding-window algorithms for FDM. Our experimental setup and results are described in Section 6. Finally, we conclude the paper in Section 7.

## 2. Related Work

Diversity maximization has been extensively studied over the last two decades. Existing studies mainly focus on two popular objectives—i.e., max–sum dispersion [11,12,13,14,15,16,29,30,31] and max–min dispersion [12,14,16,17,18,20,31], and their variants [12,32].

An early study [33] proved that both max–sum and max–min diversity maximization problems are NP-hard even in metric spaces. The classic approaches to both problems are the greedy algorithms [34,35], which achieves the best possible approximation ratio of $\frac{1}{2}$ unless P = NP. Indyk et al. [12] proposed composable coreset-based approximation algorithms for diversity maximization. Aghamolaei et al. [31] improved the approximation ratios in [12]. Ceccarello et al. [16] proposed coreset-based approximation algorithms for diversity maximization in MapReduce and streaming settings where the metric space has a bounded doubling dimension. Borassi et al. [14] proposed sliding-window algorithms for diversity maximization. Epasto et al. [36] further proposed improved sliding-window algorithms for diversity maximization specific to the Euclidean space. Drosou and Pitoura [18] studied max–min diversity maximization on dynamic data. They proposed a $\frac{b-1}{2b^{2}}$-approximation algorithm using a cover tree of base *b*. Bauckhage et al. [15] proposed an adiabatic quantum computing solution for max-sum diversification. Zhang and Gionis [19] extended diversity maximization to clustered data. Nevertheless, all the above methods only consider diversity maximization problems without fairness constraints.

There were several studies on diversity maximization under matroid constraints, of which the fairness constraints are special cases. Abbassi et al. [11] proposed a $\left(\frac{1}{2}-\varepsilon \right)$-approximation local search algorithm for max–sum diversification under matroid constraints. Borodin et al. [9] proposed a $\left(\frac{1}{2}-\varepsilon \right)$-approximation algorithm for maximizing the sum of a submodular function and a max–sum dispersion function. Cevallos et al. [30] extended the local search algorithm for distances of a negative type. They also proposed a PTAS for this problem via convex programming [29]. Bhaskara et al. [37] proposed a $\frac{1}{8}$-approximation algorithm for sum–min diversity maximization under matroid constraints using linear relaxations. Ceccarello et al. [13] proposed a coreset-based approach to matroid-constrained max–sum diversification in metric spaces of bounded doubling dimension. Nevertheless, the above methods are still not applicable to the max–min dispersion problem. The only known algorithms for fair max–min diversity maximization in [17,20,38] are offline algorithms and inefficient for data streams. We will compare our proposed algorithms with these, both theoretically and empirically. To the best of our knowledge, there has not been any previous streaming or sliding-window algorithm for fair max–min diversity maximization.

In addition to diversity maximization, fairness has also been considered in many other data summarization problems, such as *k*-center [21,22,23], determinantal point processes [8], coresets for *k*-means clustering [24,25], and submodular maximization [26,27]. However, since their optimization objectives differ from diversity maximization, the proposed algorithms for their fair variants cannot be directly used for our problem.

## 3. Preliminaries

In this section, we introduce the basic concepts and formally define the fair max–min diversity maximization problem.

Let *X* be a set of *n* elements from a metric space with distance function d(·,·) capturing the dissimilarities among elements. Recall that d(·,·) is nonnegative, symmetric, and satisfies the triangle inequality—i.e., d(x,y)+d(y,z)≥d(x,z) for any x,y,z∈X. Note that all the algorithms and analyses in this paper are general for any distance metric. We further generalize the notion of distance to an element *x* and a set *S* as the distance between *x* and its nearest neighbor in *S*—i.e., $d\left(x,S\right)=\min_{y\in S}d\left(x,y\right)$.

Our focus in this paper is to find a small subset of most diverse elements from *X*. Given a subset S⊆X, its diversity div(S) is defined as the minimum of the pairwise distances between any two distinct elements in *S*, i.e., $div\left(S\right)=\min_{x,y\in S,x\neq y}d\left(x,y\right)$. The unconstrained version of diversity maximization (DM) asks for a subset S⊆X of *k* elements maximizing div(S), i.e., $S^{*}={arg max}_{S\subseteq X:|S|=k}div\left(S\right)$. We use $\mathrm{OPT}=div(S^{*})$ to denote the diversity of the optimal solution $S^{*}$ for DM. This problem has been proven to be NP-complete [33], and no polynomial-time algorithm can achieve an approximation factor that is better than $\frac{1}{2}$ unless P=NP. One approach to DM is the $\frac{1}{2}$-approximation greedy algorithm [34,39] (known as GMM) in the offline setting.

We introduce fairness to diversity maximization when *X* is composed of several demographic groups defined by a certain sensitive attribute, e.g., sex or race. Formally, suppose that *X* is divided into *m* disjoint groups [1,⋯,m] ([m] for short) and a function c:X→[m] maps each element x∈X to its group. Let $X_{i}=\left\{x\in X:c\left(x\right)=i\right\}$ be the subset of elements from group *i* in *X*. Obviously, we have ${⋃}_{i=1}^{m}X_{i}=X$ and $X_{i}\cap X_{j}=\emptyset$ for any i≠j. The fairness constraint assigns a positive integer $k_{i}$ to each of the *m* groups and restricts the number of elements from group *i* in the solution to $k_{i}$. We assume that $\sum_{i=1}^{m}k_{i}=k$. The fair max-min diversity maximization problem is defined as:

**Definition** **1**(FDM). *Given a set X of n elements with $X={⋃}_{i=1}^{m}X_{i}$ and m size constraints $k_{1},\ldots ,k_{m}\in Z^{+}$, find a subset S that contains $k_{i}$ elements from $X_{i}$ and maximizes div(S)—i.e., $S_{f}^{*}={arg max}_{S\subseteq X\;:\;\left|S\cap X_{i}\right|=k_{i},\forall i\in \left[m\right]}div\left(S\right)$.*

We use ${\mathrm{OPT}}_{f}=div\left(S_{f}^{*}\right)$ to denote the diversity of the optimal solution $S_{f}^{*}$ for FDM. Since DM is a special case of FDM when m=1, FDM is NP-hard up to a $\frac{1}{2}$-approximation. In addition, our FDM problem is closely related to the concept of matroid [40] in combinatorics. Given a ground set *V*, a matroid is a pair M=(V,I) where I is a family of subsets of *V* (called independent sets) with the following properties: *(i)*
∅∈I; *(ii)* for each A⊆B⊆V, if B∈I then A∈I (hereditary); and *(iii)* if A∈I, B∈I, and |A|>|B|, then there exists x∈A∖B such that B∪{x}∈I (augmentation). An independent set is maximal if it is not a proper subset of any other independent set. A basic property of M is that all its maximal independent sets have the same size, denoted as the matroid’s rank. As is easy to verify, our fairness constraint is a case of rank-*k* partition matroids, where the ground set is partitioned into disjoint groups and the independent sets are exactly the sets in which, for each group, the number of elements from this group is, at most, the group capacity. Our algorithms for general *m* in Section 4 and Section 5 will be built on matroids.

In this paper, we first consider FDM in the streaming setting, where the elements in *X* arrive one at a time. Here, we use t(x) to denote the time when an element *x* is observed and $X^{\left(T\right)}=\left\{x\in X:t\left(x\right)\leq T\right\}$ to denote the subset of elements observed from *X* until time *T*. A streaming algorithm should process each element sequentially in one pass using limited space (typically independent of *n*) and return a valid approximate solution *S* (if it exists) for FDM on $X^{\left(T\right)}$ at any time *T*. We further study FDM in the sliding-window setting, where the window $W^{\left(T\right)}$ always contains the last *w* elements observed from *X* until time *T*, i.e., $W^{\left(T\right)}=\left\{x\in X:T-w+1\leq t\left(x\right)\leq T\right\}$. A sliding-window algorithm should provide a valid approximate solution *S* (if it exists) for FDM on $W^{\left(T\right)}$ at any time *T*.

## 4. Streaming Algorithms

As has been shown in Section 3, FDM is NP-hard. Thus, we focus on efficient approximation algorithms for FDM. In this section, we first describe the existing algorithms for unconstrained diversity maximization in the streaming model on which our streaming algorithms will be built. We then propose a $\frac{1-\varepsilon }{4}$-approximation streaming algorithm for FDM in the special case that there are only two groups in the dataset. Finally, we propose a $\frac{1-\varepsilon }{3m+2}$-approximation streaming algorithm for FDM on a dataset with an arbitrary number *m* of groups.

### 4.1. (Unconstrained) Streaming Algorithm

We first present the streaming algorithm of [14] for (unconstrained) diversity maximization in Algorithm 1. Let $d_{min}=\min_{x,y\in X,x\neq y}$d(x,y), $d_{max}=\max_{x,y\in X,x\neq y}d\left(x,y\right)$ and $\Delta =\frac{d_{max}}{d_{min}}$. Obviously, it always holds that $\mathrm{OPT}\in [d_{min},d_{max}]$. First, it maintains a sequence U of values for guessing OPT within a relative error of 1−ε and initializes an empty solution $S_{\mu }$ for each μ∈U before processing the stream (Lines 1 and 2). Then, for each x∈X and each μ∈U, if $S_{\mu }$ contains less than *k* elements and the distance between *x* and $S_{\mu }$ is at least μ, it will add *x* to $S_{\mu }$ (Lines 3–6). After processing all elements in *X*, the candidate solution that contains *k* elements and maximizes the diversity is returned as the solution *S* for DM (Line 7). Algorithm 1 is proven to be a $\frac{1-\varepsilon }{5}$-approximation algorithm for max–min diversity maximization [14]. In Theorem 1, its approximation ratio is improved to $\frac{1-\varepsilon }{2}$ by refining the analysis of [14].
**Algorithm 1** SDM**Input**:Stream *X*, distance metric d(·,·), parameter ε∈(0,1), solution size $k\in Z^{+}$**Output**:A set S⊆X with |S|=k1:$U=\{\frac{d_{min}}{{\left(1-\varepsilon \right)}^{j}}\;:\;j\in Z_{0}^{+}\wedge {\left(1-\varepsilon \right)}^{j}\geq \frac{d_{min}}{d_{max}}\}$2:Initialize $S_{\mu }=\emptyset$ for each μ∈U3:**for all** x∈X **do**4:    **for all** μ∈U **do**5:        **if** $|S_{\mu }|<k$ and $d(x,S_{\mu })\geq \mu$ **then**6:           $S_{\mu }\leftarrow S_{\mu }\cup \left\{x\right\}$7:**return** $S\leftarrow {arg max}_{\mu \in U:|S_{\mu }|=k}div\left(S_{\mu }\right)$

**Theorem** **1**.
*Algorithm 1 is a $\frac{1-\varepsilon }{2}$-approximation algorithm for max–min diversity maximization.*


**Proof.** For each μ∈U, there are two cases for $S_{\mu }$ after processing all elements in *X*: (1) If $|S_{\mu }|=k$, the condition of Line 5 guarantees that $div(S_{\mu })\geq \mu$; (2) If $|S_{\mu }|<k$, it holds that $d(x,S_{\mu })<\mu$ for every $x\in X\setminus S_{\mu }$ since the fact that *x* is not added to $S_{\mu }$ implies that $d(x,S_{\mu })<\mu$, as $|S_{\mu }|<k$. Let us consider a candidate solution $S_{\mu }$ with $|S_{\mu }|<k$. Suppose that $S^{*}=\left\{s_{1}^{*},\ldots ,s_{k}^{*}\right\}$ is the optimal solution for DM on *X*. We define a function $f:S^{*}\to S_{\mu }$ that maps each element in $S^{*}$ to its nearest neighbor in $S_{\mu }$. As is shown above, $d(s^{*},f\left(s^{*}\right))<\mu$ for each $s^{*}\in S^{*}$. Because $|S_{\mu }|<k$ and $|S^{*}|=k$, two distinct elements $s_{a}^{*},s_{b}^{*}\in S^{*}$ with $f\left(s_{a}^{*}\right)=f\left(s_{b}^{*}\right)$ must exist. For such $s_{a}^{*},s_{b}^{*}$, we have
$$d\left(s_{a}^{*},s_{b}^{*}\right)\leq d\left(s_{a}^{*},f\left(s_{a}^{*}\right)\right)+d\left(s_{b}^{*},f\left(s_{b}^{*}\right)\right)<2\mu$$
according to the triangle inequality. Thus, $\mathrm{OPT}=div\left(S^{*}\right)\leq d\left(s_{a}^{*},s_{b}^{*}\right)<2\mu$ if $|S_{\mu }|<k$. Let $\mu^{\prime }$ be the smallest μ∈U with $|S_{\mu }|<k$. We obtained $div\left(S^{*}\right)<2\mu^{\prime }$ from the above results. Additionally, for $\mu^{\prime \prime }=\left(1-\varepsilon \right)\mu^{\prime }$, we must have $|S_{\mu^{\prime \prime }}|=k$ and $div\left(S_{\mu^{\prime \prime }}\right)\geq \mu^{\prime \prime }$. Therefore, we have $div\left(S\right)\geq \mu^{\prime \prime }=\left(1-\varepsilon \right)\mu^{\prime }\geq \frac{1-\varepsilon }{2}\cdot div\left(S^{*}\right)$.    □

In terms of complexity, Algorithm 1 stores $O(\frac{k\log\Delta }{\varepsilon })$ elements and takes $O(\frac{k\log\Delta }{\varepsilon })$ time per element, since it makes $O(\frac{\log\Delta }{\varepsilon })$ guesses for OPT, keeps, at most, *k* elements in each candidate and requires, at most, *k* distance computations to decide whether to add an element to a candidate.

### 4.2. Fair Streaming Algorithm for m=2

The procedure of our streaming algorithm in case of m=2, called SFDM1, is described in Algorithm 2 and illustrated in Figure 3. In general, the algorithm runs in two phases: stream processing and post-processing. In the stream processing (Lines 1–6), for each guess μ∈U of ${\mathrm{OPT}}_{f}$, it utilizes Algorithm 1 to keep a group-blind candidate $S_{\mu }$ with size constraint *k* and two group-specific candidates $S_{\mu ,1}$ and $S_{\mu ,2}$ with size constraints $k_{1}$ and $k_{2}$ for $X_{1}$ and $X_{2}$, respectively. The only difference from Algorithm 1 is that the elements are filtered by group to maintain $S_{\mu ,1}$ and $S_{\mu ,2}$. After processing all elements of *X* in one pass, it will post-process the group-blind candidates to make them satisfy the fairness constraint (Lines 7–15). The post-processing is only performed on a subset $U^{\prime }$ of U, where $S_{\mu }$ contains *k* elements and $S_{\mu ,i}$ contains $k_{i}$ elements for each group i∈{1,2}. For each $\mu \in U^{\prime }$, $S_{\mu }$, either has satisfied the fairness constraint or has one over-filled group $i_{o}$ and another under-filled group $i_{u}$. If $S_{\mu }$ is not yet a fair solution, $S_{\mu }$ will be balanced for fairness by first adding $k_{i_{u}}-k_{i_{u}}^{\prime }$ elements, where $k_{i_{u}}^{\prime }=\left|S_{\mu }\cap X_{i_{u}}\right|$, from $S_{\mu ,i_{u}}$ to $S_{\mu }$, and then removing the same number of elements from $S_{\mu }\cap X_{i_{o}}$. The elements to be added and removed are selected greedily, as in GMM [39], to minimize the loss in diversity: the element in $S_{\mu ,i_{u}}$ that is the furthest from $S_{\mu }\cap X_{i_{u}}$ is picked for each insertion; and the element in $S_{\mu }\cap X_{i_{o}}$ that is the closest to $S_{\mu }\cap X_{i_{u}}$ is picked for each deletion. Finally, the fair candidate with the maximum diversity after post-processing is returned as the final solution for FDM (Line 16). Next, we will theoretically analyze the approximation ratio and complexity of SFDM1.
**Algorithm 2** SFDM1**Input**:Stream $X=X_{1}\cup X_{2}$, distance metric d(·,·), parameter ε∈(0,1), size constraints $k_{1},k_{2}\in Z^{+}$ ($k=k_{1}+k_{2}$)**Output**:A set S⊆X s.t. $\left|S\cap X_{i}\right|=k_{i}$ for i∈{1,2}▹Stream processing1:$U=\{\frac{d_{min}}{{\left(1-\varepsilon \right)}^{j}}\;:\;j\in Z_{0}^{+}\wedge {\left(1-\varepsilon \right)}^{j}\geq \frac{d_{min}}{d_{max}}\}$2:Initialize $S_{\mu },S_{\mu ,i}=\emptyset$ for every μ∈U and i∈{1,2}3:**for all** x∈X **do**4:    Run Lines 3–6 of Algorithm 1 to update $S_{\mu }$ w.r.t. *x*5:    **if** c(x)=i **then**6:        Run Lines 3–6 of Algorithm 1 to update $S_{\mu ,i}$ w.r.t. *x* with size constraint $k_{i}$▹Post-processing7:$U^{\prime }=\left\{\mu \in U:\left|S_{\mu }\right|=k\wedge \left|S_{\mu ,i}\right|=k_{i},\forall i\in \left\{1,2\right\}\right\}$8:**for all** $\mu \in U^{\prime }$ **do**9:    **if** $\left|S_{\mu }\cap X_{i}\right|<k_{i}$ for some i∈{1,2} **then**10:        **while** $\left|S_{\mu }\cap X_{i}\right|<k_{i}$ **do**11:           $x^{+}\leftarrow {arg max}_{x\in S_{\mu ,i}}d\left(x,S_{\mu }\cap X_{i}\right)$12:           $S_{\mu }\leftarrow S_{\mu }\cup \left\{x^{+}\right\}$13:        **while** $|S_{\mu }|>k$ **do**14:           $x^{-}\leftarrow {arg min}_{x\in S_{\mu }\setminus X_{i}}d\left(x,S_{\mu }\cap X_{i}\right)$15:           $S_{\mu }\leftarrow S_{\mu }\setminus \left\{x^{-}\right\}$16:**return** $S\leftarrow {arg max}_{\mu \in U^{\prime }}div\left(S_{\mu }\right)$

**Theoretical Analysis:** We prove that SFDM1 achieves an approximation ratio of $\frac{1-\varepsilon }{4}$ for FDM, where ε∈(0,1), in Theorem 2. The proof is based on *(i)* the existence of $\mu^{\prime }\in U^{\prime }$ such that $\mu^{\prime }\geq \frac{1-\varepsilon }{2}\cdot {\mathrm{OPT}}_{f}$ (Lemma 1) and *(ii)* $div\left(S_{\mu }\right)\geq \frac{\mu }{2}$ for each $\mu \in U^{\prime }$ after post-processing (Lemma 2). Then, we analyze the complexity of SFDM1 in Theorem 3.

**Lemma** **1**.
*Let $\mu^{\prime }$ be the largest $\mu \in U^{\prime }$. It holds that $\mu^{\prime }\geq \frac{1-\varepsilon }{2}\cdot {\mathrm{OPT}}_{f}$, where ${\mathrm{OPT}}_{f}$ is the optimal diversity of FDM on X.*


**Proof.** First of all, we have ${\mathrm{OPT}}_{f}\leq \mathrm{OPT}$, where OPT is the optimal diversity of unconstrained DM with $k=k_{1}+k_{2}$ on *X*, since any valid solution for FDM must also be a valid solution for DM. Moreover, it holds that ${\mathrm{OPT}}_{f}\leq {\mathrm{OPT}}_{k_{i}}$, where ${\mathrm{OPT}}_{k_{i}}$ is the optimal diversity of unconstrained DM with size constraint $k_{i}$ on $X_{i}$ for both i∈{1,2}, because the optimal solution must contain $k_{i}$ elements from $X_{i}$ and div(·) is a monotonically non-increasing function—i.e., div(S∪{x})≤div(S) for any S⊆X and x∈S∖X. Therefore, we prove that ${\mathrm{OPT}}_{f}\leq div\left(S^{*}\cap X_{i}\right)\leq {\mathrm{OPT}}_{k_{i}}$.Then, according to the results of Theorem 1, we have OPT<2μ if $S_{\mu }<k$ and ${\mathrm{OPT}}_{k_{i}}<2\mu$ if $S_{\mu ,i}<k_{i}$ for each i∈{1,2}. Note that $\mu^{\prime }$ is the largest μ∈U, such that $|S_{\mu }|=k$, $\left|S_{\mu ,1}\right|=k_{1}$, and $\left|S_{\mu ,2}\right|=k_{2}$ after stream processing. For $\mu^{\prime \prime }=\frac{\mu^{\prime }}{1-\varepsilon }\in U$, we have either $|S_{\mu^{\prime \prime }}|<k$ or $\left|S_{\mu^{\prime \prime },i}\right|<k_{i}$ for some i∈{1,2}. Therefore, it holds that ${\mathrm{OPT}}_{f}<2\mu^{\prime \prime }\leq \frac{2}{1-\varepsilon }\cdot \mu^{\prime }$ and we conclude the proof.    □

**Lemma** **2**.
*For each $\mu \in U^{\prime }$, the candidate solution $S_{\mu }$ must satisfy $div\left(S_{\mu }\right)\geq \frac{\mu }{2}$ and $\left|S_{\mu }\cap X_{i}\right|=k_{i}$ for both i∈{1,2} after post-processing.*


**Proof.** The candidate $S_{\mu }$ before post-processing has exactly $k=k_{1}+k_{2}$ elements but may not contain $k_{1}$ elements from $X_{1}$ and $k_{2}$ elements from $X_{2}$. If $S_{\mu }$ has exactly $k_{1}$ elements from $X_{1}$ and $k_{2}$ elements from $X_{2}$, and thus the post-processing is skipped, we have $div(S_{\mu })\geq \mu$ according to Theorem 1. Otherwise, assuming that $\left|S_{\mu }\cap X_{1}\right|=k_{1}^{\prime }<k_{1}$, we will add $k_{1}-k_{1}^{\prime }$ elements from $S_{\mu ,1}$ to $S_{\mu }$ and remove $k_{1}-k_{1}^{\prime }$ elements from $S_{\mu }\cap X_{2}$ to ensure the fairness constraint. In Line 16, all the $k_{1}$ elements in $S_{\mu ,1}$ can be selected for insertion. Since the minimum distance between any pair of elements in $S_{\mu ,1}$ is at least μ, we can find, at most, one element $x\in S_{\mu ,1}$, such that $d\left(x,y\right)<\frac{\mu }{2}$ for each $y\in S_{\mu }\cap X_{1}$. This means that there are at least $k_{1}-k_{1}^{\prime }$ elements from $S_{\mu ,1}$ whose distances to all the existing elements in $S_{\mu }\cap X_{1}$ are greater than $\frac{\mu }{2}$. Accordingly, after adding $k_{1}-k_{1}^{\prime }$ elements from $S_{\mu ,1}$ to $S_{\mu }$ greedily, it still holds that $d\left(x,y\right)\geq \frac{\mu }{2}$ for any $x,y\in S_{\mu }\cap X_{1}$. In Line 14, for each element $x\in S_{\mu }\cap X_{2}$, there is, at most, one (newly added) element $y\in S_{\mu }\cap X_{1}$ such that $d\left(x,y\right)<\frac{\mu }{2}$. Meanwhile, it is guaranteed that *y* is the nearest neighbor of *x* in $S_{\mu }$ in this case. Therefore, in Line 14, every $x\in S_{\mu }\cap X_{2}$ with $d\left(x,S_{\mu }\cap X_{2}\right)<\frac{\mu^{\prime }}{2}$ is removed, since there are, at most, $k_{1}-k_{1}^{\prime }$ such elements and the one with the smallest $d(x,S_{\mu }\cap X_{2})$ is removed at each step. Therefore, $S_{\mu }$ contains $k_{1}$ elements from $X_{1}$ and $k_{2}$ elements from $X_{2}$ and $div\left(S_{\mu }\right)\geq \frac{\mu }{2}$ after post-processing.    □

**Theorem** **2**.
SFDM1
*returns a $\frac{1-\varepsilon }{4}$-approximate solution for FDM.*


**Proof.** According to the results of Lemmas 1 and 2, we have $div\left(S\right)\geq div\left(S_{\mu^{\prime }}\right)\geq \frac{\mu^{\prime }}{2}\geq \frac{1-\varepsilon }{4}\cdot {\mathrm{OPT}}_{f}$, where $\mu^{\prime }=\max_{\mu \in U^{\prime }}\mu$.    □

**Theorem** **3**.
SFDM1
*stores $O(\frac{k\log\Delta }{\varepsilon })$ elements in memory, takes $O(\frac{k\log\Delta }{\varepsilon })$ time per element for stream processing, and $O(\frac{k^{2}\log\Delta }{\varepsilon })$ time for post-processing.*


**Proof.** SFDM1 keeps three candidates for each μ∈U and O(k) elements in each candidate. Hence, the total number of stored elements is $O(\frac{k\log\Delta }{\varepsilon })$, since $\left|U\right|=O\left(\frac{\log\Delta }{\varepsilon }\right)$. The stream processing performs, at most, $O(\frac{k\log\Delta }{\varepsilon })$ distance computations per element. Finally, for each $\mu \in U^{\prime }$ in the post-processing, at most $k_{i}\left(k_{i}-k_{i}^{\prime }\right)$ distance computations are performed to select the elements in $S_{\mu ,i}$ that are to be added to $S_{\mu }$. To find the elements that are to be removed, at most $k(k_{i}-k_{i}^{\prime })$ distance computations are needed. Thus, the time complexity for post-processing is $O(\frac{k^{2}\log\Delta }{\varepsilon })$ as $\left|U^{\prime }\right|=O\left(\frac{\log\Delta }{\varepsilon }\right)$.    □

**Comparison with Prior Art:** The idea of finding a solution and balancing it for fairness in SFDM1 has also been used for FairSwap [17]. However, FairSwap only works in the offline setting, which keeps the dataset in memory and requires random accesses for computation, whereas SFDM1 works in the streaming setting, which scans the dataset in one pass and uses only the elements in the candidates for post-processing. Compared with FairSwap, SFDM1 reduces the space complexity from O(n) to $O(\frac{k\log\Delta }{\varepsilon })$ and the time complexity from O(nk) to $O(\frac{k^{2}\log\Delta }{\varepsilon })$ at the expense of lowering the approximation ratio by a factor of 1−ε.

### 4.3. Fair Streaming Algorithm for General *m*

The detailed procedure of our streaming algorithm, which can work with an arbitrary m≥2, called SFDM2, is presented in Algorithm 3. Similar to SFDM1, it also has two phases: stream processing and post-processing. In the stream processing (Lines 1–7), it utilizes Algorithm 1 to keep a group-blind candidate $S_{\mu }$ and *m* group-specific candidates $S_{\mu ,1},\ldots ,S_{\mu ,m}$ for all the *m* groups. The difference from SFDM1 is that the size constraint of each group-specific candidate for each group *i* is *k* instead of $k_{i}$. Then, after processing all elements in *X*, a post-processing scheme is required to ensure the fairness of candidates. Nevertheless, the post-processing procedures are totally different from SFDM1, since the swap-based balancing strategy cannot guarantee the validity of the solution with any theoretical bound. Like SFDM1, the post-processing is performed on a subset $U^{\prime }$, where $S_{\mu }$ has *k* elements and $S_{\mu ,i}$ has at least $k_{i}$ elements for each group *i* (Line 8). For each $\mu \in U^{\prime }$, it initializes with a subset $S_{\mu }^{\prime }$ of $S_{\mu }$ (Line 10). For an over-filled group *i*, i.e., $\left|S_{\mu }\cap X_{i}\right|>k_{i}$, $S_{\mu }^{\prime }$ contains $k_{i}$ arbitrary elements from $S_{\mu }$. For an under-filled or exactly filled group *i*, i.e., $\left|S_{\mu }\cap X_{i}\right|\leq k_{i}$, $S_{\mu }^{\prime }$ contains all $k_{i}^{\prime }=\left|S_{\mu }\cap X_{i}\right|$ elements from $S_{\mu }$. Next, new elements from under-filled groups should be added to $S_{\mu }^{\prime }$ so that $S_{\mu }^{\prime }$ is a fair solution. The method used to find the elements that are to be added is to divide the set $S_{all}$ of elements in all candidates into a set C of clusters, which guarantees that $d\left(x,y\right)\geq \frac{\mu }{m+1}$ for any $x\in C_{a}$ and $y\in C_{b}$ (Lines 12–15), where $C_{a}$ and $C_{b}$ are two different clusters in C. Then, $S_{\mu }^{\prime }$ is limited to contain, at most, one element from each cluster after new elements are added so that $div\left(S_{\mu }^{\prime }\right)\geq \frac{\mu }{m+1}$. Meanwhile, $S_{\mu }^{\prime }$ should still satisfy the fairness constraint. To meet both requirements, the problem of adding new elements to $S_{\mu }^{\prime }$ is formulated as an instance of matroid intersection [41,42,43], as will be discussed subsequently (Line 17). Finally, it returns $S_{\mu }^{\prime }$ containing *k* elements with maximum diversity after post-processing as the final solution for FDM (Line 18). An illustration of the post-processing procedure of SFDM2 is given in Figure 4.
**Algorithm 3** SFDM2**Input**:Stream $X={⋃}_{i=1}^{m}X_{i}$, distance metric *d*, parameter ε∈(0,1), size constraints $k_{1},\ldots ,k_{m}\in Z^{+}$ ($k=\sum_{i=1}^{m}k_{i}$)**Output**:A set S⊆X s.t. $\left|S\cap X_{i}\right|=k_{i}$, ∀i∈[m]▹Stream processing1:$U=\{\frac{d_{min}}{{\left(1-\varepsilon \right)}^{j}}\;:\;j\in Z_{0}^{+}\wedge {\left(1-\varepsilon \right)}^{j}\geq \frac{d_{min}}{d_{max}}\}$2:Initialize $S_{\mu },S_{\mu ,i}=\emptyset$ for every μ∈U and i∈[m]3:**for all** x∈X **do**4:    **for all** μ∈U and i∈[m] **do**5:        Run Lines 3–6 of Algorithm 1 to update $S_{\mu }$ w.r.t. *x*6:        **if** c(x)=i **then**7:           Run Lines 3–6 of Algorithm 1 to update $S_{\mu ,i}$ w.r.t. *x*▹Post-processing8:$U^{\prime }=\left\{\mu \in U:\left|S_{\mu }\right|=k\wedge \left|S_{\mu ,i}\right|\geq k_{i},\forall i\in \left[m\right]\right\}$9:**for all** $\mu \in U^{\prime }$ **do**10:    For each group i∈[m], pick $\min(k_{i},\left|S_{\mu }\cap X_{i}\right|)$ elements arbitrarily from $S_{\mu }$ as $S_{\mu }^{\prime }$11:    Let $S_{all}=\left({⋃}_{i=1}^{m}S_{\mu ,i}\right)\cup S_{\mu }$ and $l=|S_{all}|$12:    Create *l* clusters $C=\{C_{1},\ldots ,C_{l}\}$, each of which contains one element in $S_{all}$13:    **while** there exist $C_{a},C_{b}\in C$ s.t. $d\left(x,y\right)<\frac{\mu }{m+1}$ for some $x\in C_{a}$ and $y\in C_{b}$ **do**14:        Merge $C_{a},C_{b}$ into a new cluster $C^{\prime }=C_{a}\cup C_{b}$15:        $C\leftarrow C\setminus \left\{C_{a},C_{b}\right\}\cup \left\{C^{\prime }\right\}$16:    Let $M_{1}=\left(S_{all},I_{1}\right)$ and $M_{2}=\left(S_{all},I_{2}\right)$ be two matroids, where $S\in I_{1}$ iff $\left|S\cap X_{i}\right|\leq k_{i}$, ∀i∈[m] and $S\in I_{2}$ iff |S∩C|≤1, ∀C∈C17:    Run Algorithm 4 to augment $S_{\mu }^{\prime }$ such that $S_{\mu }^{\prime }$ is a maximum cardinality set in $I_{1}\cap I_{2}$18:**return** $S\leftarrow {arg max}_{\mu \in U^{\prime }:\left|S_{\mu }^{\prime }\right|=k}div\left(S_{\mu }^{\prime }\right)$

**Matroid Intersection:** Next, we describe how to use matroid intersection for solution augmentation in SFDM2. We define the first rank-*k* matroid $M_{1}=\left(V,I_{1}\right)$ based on the fairness constraint, where the ground set *V* is $S_{all}$ and $S\in I_{1}$ iff $\left|S\cap X_{i}\right|\leq k_{i}$, ∀i∈[m]. Intuitively, a set *S* is fair if it is a maximal independent set in $I_{1}$. Moreover, we define the second rank-*l* (l=|C|) matroid $M_{2}=\left(V,I_{2}\right)$ on the set C of clusters, where the ground set *V* is also $S_{all}$ and $S\in I_{2}$ if |S∩C|≤1, ∀C∈C. Accordingly, the problem of adding new elements to $S_{\mu }^{\prime }$ to ensure fairness is an instance of the matroid intersection problem, which aims to find a maximum cardinality set $S\in I_{1}\cap I_{2}$ for $M_{1}=\left(S_{all},I_{1}\right)$ and $M_{2}=\left(S_{all},I_{2}\right)$. Here, we adopt Cunningham’s algorithm [41], a well-known solution for the matroid intersection problem based on the augmentation graph in Definition 2.

**Definition** **2**(Augmentation Graph [41]). *Given two matroids $M_{1}=\left(V,I_{1}\right)$ and $M_{2}=\left(V,I_{2}\right)$, a set S⊂V, such that $S\in I_{1}\cap I_{2}$, and two sets $V_{1}=\left\{x\in V\setminus S:S\cup \left\{x\right\}\in I_{1}\right\}$ and $V_{2}=\left\{x\in V\setminus S:S\cup \left\{x\right\}\in I_{2}\right\}$, an augmentation graph is a digraph G=(V∪{a,b},E), where a,b∉V. There is an edge (a,x)∈E for each $x\in V_{1}$. There is an edge (x,b)∈E for each $x\in V_{2}$. There is an edge (y,x)∈E for each x∈V∖S, y∈S, such that $S\cup \left\{x\right\}\notin I_{1}$ and $S\cup \left\{x\right\}\setminus \left\{y\right\}\in I_{1}$. There is an edge (x,y)∈E for each x∈V∖S, y∈S, such that $S\cup \left\{x\right\}\notin I_{2}$ and $S\cup \left\{x\right\}\setminus \left\{y\right\}\in I_{2}$.*

Specifically, the Cunningham’s algorithm [41] is initialized with S=∅ (or any $S\in I_{1}\cap I_{2}$). At each step, it builds an augmentation graph *G* for $M_{1}$, $M_{2}$, and *S*. If there is no directed path from *a* to *b* in *G*, then *S* is already a maximum cardinality set. Otherwise, it finds the shortest path $P^{*}$ from *a* to *b* in *G*, and augments *S* according to $P^{*}$. For each x∈V∖S, except *a* and *b*, add *x* to *S*; for each x∈S, remove *x* from *S*. We adapt Cunningham’s algorithm [41] for our problem, as shown in Algorithm 4. Our algorithm is initialized with $S_{\mu }^{\prime }$ instead of *∅*. In addition, to reduce the cost of building *G* and maximize the diversity, it first adds the elements in $V_{1}\cap V_{2}$ greedily to $S_{\mu }^{\prime }$ until $V_{1}\cap V_{2}=\emptyset$. This is because a shortest path, $P^{*}=\langle a,x,b\rangle$ in *G*, exists for any $x\in V_{1}\cap V_{2}$, which is easy to verify from Definition 2. Finally, if |S|<k after the above procedures, the standard Cunningham’s algorithm will be used to augment *S* to ensure the maximality of *S*.
**Algorithm 4** Matroid Intersection**Input**:Two matroids $M_{1}=\left(V,I_{1}\right)$, $M_{2}=\left(V,I_{2}\right)$, distance metric *d*, initial set $S_{0}\subseteq V$**Output**:A maximum cardinality set S⊆V in $I_{1}\cap I_{2}$1:Initialize $S\leftarrow S_{0}$, $V_{1}=\left\{x\in V\setminus S:S\cup \left\{x\right\}\in I_{1}\right\}$, and $V_{2}=\left\{x\in V\setminus S:S\cup \left\{x\right\}\in I_{2}\right\}$2:**while** $V_{1}\cap V_{2}\neq \emptyset$ **do**3:    $x^{*}\leftarrow {arg max}_{x\in V_{1}\cap V_{2}}d\left(x,S\right)$ and $S\leftarrow S\cup \{x^{*}\}$4:    **for all** $x\in V_{1}$ **do**5:        $V_{1}\leftarrow V_{1}\setminus \left\{x\right\}$ if $S\cup \left\{x\right\}\notin I_{1}$6:    **for all** $x\in V_{2}$ **do**7:        $V_{2}\leftarrow V_{2}\setminus \left\{x\right\}$ if $S\cup \left\{x\right\}\notin I_{2}$8:Build an augmentation graph *G* for *S*9:**while** there is a directed path from *a* to *b* in *G* **do**10:    Let $P^{*}$ be a shortest path from *a* to *b* in *G*11:    **for all** $x\in P^{*}\setminus \left\{a,b\right\}$ **do**12:        S←S∪{x} if x∉S13:        S←S∖{x} if x∈S14:    Rebuild *G* for the updated *S*15:**return***S*

**Theoretical Analysis:** We prove that SFDM2 achieves an approximation ratio of $\frac{1-\varepsilon }{3m+2}$ for FDM. The high-level idea of the proof is to connect the clustering procedure in post-processing with the notion of matroid and then to utilize the geometric properties of the clusters and the theoretical results of matroid intersection for approximation. Next, we first show that the set C of clusters has several important properties (Lemma 3). Then, we prove that Algorithm 4 can return a fair solution for a specific μ based on the properties of C (Lemma 4). Finally, we analyze the time and space complexities of SFDM2 in Theorem 5.

**Lemma** **3**.
*The set C of clusters has the following properties: (i) for any $x\in C_{a}$ and $y\in C_{b}$ (a≠b), $d\left(x,y\right)\geq \frac{\mu }{m+1}$; (ii) each cluster C contains, at most, one element from $S_{\mu }$ and $S_{\mu ,i}$ for any i∈[m]; (iii) for any x,y∈C, $d\left(x,y\right)<\frac{m}{m+1}\cdot \mu$.*


**Proof.** First of all, Property (i) holds from Lines 12–15 of Algorithm 3, since all clusters that do not satisfy it have been merged. Then, we prove Property (ii) by contradiction. Let us construct an undirected graph G=(V,E) for a cluster C∈C, where *V* is the set of elements in *C* and there exists an edge (x,y)∈E iff $d\left(x,y\right)<\frac{\mu }{m}$. Based on Algorithm 3, for any x∈C, there must exist some y∈C (x≠y) such that $d\left(x,y\right)<\frac{\mu }{m}$. Therefore, *G* is a connected graph. Suppose that *C* contains more than one element from $S_{\mu }$ or $S_{\mu ,i}$ for some i∈[m]. Let $P_{x,y}=\left(x,\ldots ,y\right)$ be the shortest path of *G* between *x* and *y*, where *x* and *y* are both from $S_{\mu }$ or $S_{\mu ,i}$. Next, we show that the length of $P_{x,y}$ is, at most, m+1. If the length of $P_{x,y}$ is longer than m+1, there will be a sub-path $P_{x^{\prime },y^{\prime }}$ of $P_{x,y}$ where $x^{\prime }$ and $y^{\prime }$ are both from $S_{\mu }$ or $S_{\mu ,i}$, and this violates the fact that $P_{x,y}$ is the shortest. Since the length of $P_{x,y}$ is, at most, m+1, we have $d\left(x,y\right)<\left(m+1\right)\cdot \frac{\mu }{m+1}=\mu$, which contradicts the fact that d(x,y)≥μ, as they are both from $S_{\mu }$ or $S_{\mu ,i}$. Finally, Property (iii) is a natural extension of Property (ii): since each cluster *C* contains, at most, one element from $S_{\mu }$ and $S_{\mu ,i}$ for any i∈[m], *C* has, at most, m+1 elements. Therefore, for any two elements x,y∈C, the length of the path between them is, at most, *m* in *G* and $d\left(x,y\right)<m\cdot \frac{\mu }{m+1}=\frac{m}{m+1}\cdot \mu$.    □

**Lemma** **4**.
*If ${\mathrm{OPT}}_{f}\geq \frac{3m+2}{m+1}\cdot \mu$, then Algorithm 4 returns a size-k subset $S_{\mu }^{\prime }$, such that $S_{\mu }^{\prime }\in I_{1}\cap I_{2}$ and $div\left(S_{\mu }^{\prime }\right)\geq \frac{\mu }{m+1}$.*


**Proof**. First of all, the initial $S_{\mu }^{\prime }$ is a subset of $S_{\mu }$. According to Property (ii) of Lemma 3, all elements of $S_{\mu }^{\prime }$ are in different clusters of C, and thus $S_{\mu }^{\prime }\in I_{1}\cap I_{2}$. The theoretical results in [41] guarantee that Algorithm 4 can find a size-*k* set in $I_{1}\cap I_{2}$, as long as it exists. Next, we will show such a set exists when ${\mathrm{OPT}}_{f}\geq \frac{3m+2}{m+1}\cdot \mu$. To verify this, we need to identify $k_{i}$ clusters of C that contain at least one element from $X_{i}$ for each i∈[m] and show that all $k=\sum_{i=1}^{m}k_{i}$ clusters are distinct. Here, we consider two cases for each group i∈[m].
**Case 1:** For each i∈[m], such that $k_{i}\leq \left|S_{\mu ,i}\right|<k$, we have $d(x,S_{\mu ,i})<\mu$ for each $x\in X_{i}$. Given the optimal solution $S_{f}^{*}$, we define a function *f* that maps each $x^{*}\in S_{f}^{*}$ to its nearest neighbor in $S_{\mu ,i}$. For two elements $x_{a}^{*},x_{b}^{*}\in S_{f}^{*}$ in these groups, we have $d(x_{a}^{*},f\left(x_{a}^{*}\right))<\mu$, $d(x_{b}^{*},f\left(x_{b}^{*}\right))<\mu$, and $d\left(x_{a}^{*},x_{b}^{*}\right)\geq {\mathrm{OPT}}_{f}=div\left(S_{f}^{*}\right)$. Therefore, $d\left(f\left(x_{a}^{*}\right),f\left(x_{b}^{*}\right)\right)>{\mathrm{OPT}}_{f}-2\mu$. Since ${\mathrm{OPT}}_{f}\geq \frac{3m+2}{m+1}\cdot \mu$, $d\left(f\left(x_{a}^{*}\right),f\left(x_{b}^{*}\right)\right)>\frac{3m+2}{m+1}\cdot \mu -2\mu =\frac{m}{m+1}\cdot \mu$. According to Property (iii) of Lemma 3, it is guaranteed that $f(x_{a}^{*})$ and $f(x_{b}^{*})$ are in different clusters. By identifying all the clusters that contain $f(x^{*})$ for all $x^{*}\in S_{f}^{*}$, we found $k_{i}$ clusters for each group i∈[m] such that $k_{i}\leq \left|S_{\mu ,i}\right|<k$. All the clusters that were found are guaranteed to be distinct.**Case 2:** For all i∈[m] such that $|S_{\mu ,i}|=k$, we are able to find *k* clusters that contain one element from $S_{\mu ,i}$ based on Property (ii) of Lemma 3. For such a group *i*, even though $k-k_{i}$ clusters have been identified for all other groups, there are still at least $k_{i}$ clusters available for selection. Therefore, we can always find $k_{i}$ clusters that are distinct from all the clusters identified by any other group for such a group $X_{i}$.
Considering both cases, we have proven the existence of a size-*k* set in $I_{1}\cap I_{2}$. Finally, for any set $S\in I_{2}$, we have $div\left(S\right)\geq \frac{\mu }{m+1}$ according to Property (i) of Lemma 3.    □

**Theorem** **4**.
SFDM2
*is a $\frac{1-\varepsilon }{3m+2}$-approximation algorithm for FDM.*


**Proof.** Let $\mu^{-}$ be the smallest $\mu \notin U^{\prime }$. It holds that $\mu^{-}\geq \frac{{\mathrm{OPT}}_{f}}{2}$ (see Lemma 1). Thus, there is some $\mu <\mu^{-}$ in $U^{\prime }$, such that $\mu \in [\frac{\left(m+1)(1-\varepsilon \right)}{3m+2}\cdot {\mathrm{OPT}}_{f},\frac{m+1}{3m+2}\cdot {\mathrm{OPT}}_{f}]$, as $\frac{m+1}{3m+2}<\frac{1}{2}$ for any $m\in Z^{+}$. Therefore, SFDM2 provides a fair solution *S*, such that $div\left(S\right)\geq div\left(S_{\mu }^{\prime }\right)\geq \frac{\mu }{m+1}\geq \frac{1-\varepsilon }{3m+2}\cdot {\mathrm{OPT}}_{f}$.    □

**Theorem** **5**.
SFDM2
*keeps $O(\frac{km\log\Delta }{\varepsilon })$ elements in memory, takes $O(\frac{k\log\Delta }{\varepsilon })$ time per element in the stream processing, and spends $O\left(\frac{k^{2}m\log\Delta }{\varepsilon }\cdot \left(m+\log^{2}k\right)\right)$ time for post-processing.*


**Proof.** SFDM2 keeps m+1 candidates for each μ∈U and O(k) elements in each candidate. So, the total number of elements stored by SFDM2 is $O(\frac{km\log\Delta }{\varepsilon })$. Only two candidates are checked in streaming processing for each element and thus $O(\frac{k\log\Delta }{\varepsilon })$ distance computations are needed. In the post-processing of each μ, we need O(k) time to get the initial solution, $O(k^{2}m^{2})$ time to cluster $S_{all}$, and $O(k^{2}m)$ time to augment the candidate using Lines 2–7 of Algorithm 4. The time complexity of Cunningham’s algorithm is $O(k^{2}m\log^{2}k)$ according to [42,43]. In sum, the overall time complexity of post-processing is $O\left(\frac{k^{2}m\log\Delta }{\varepsilon }\cdot \left(m+\log^{2}k\right)\right)$.    □

**Comparison with Prior Art:** Existing methods have aimed to find a fair solution based on matroid intersection for fair *k*-center [21,22,44] and fair max–min diversity maximization [17]. SFDM2 adopts a similar method to FairFlow [17] to construct the clusters and matroids. However, FairFlow solves matroid intersection as a max-flow problem on a directed graph. Its solution is of poor quality in practice, particularly when *m* is large. Therefore, SFDM2 uses a different method for matroid intersection based on Cunningham’s algorithm, which initializes with a partial solution instead of an empty set for higher efficiency and adds elements greedily like GMM [39] for higher diversity. Hence, SFDM2 has a significantly higher solution quality than FairFlow in practice, though it has a slightly lower approximation ratio.

## 5. Sliding-Window Algorithms

In this section, we extend our streaming algorithms, i.e., SFDM1 and SFDM2, to the sliding-window model. In Section 5.1, we first present the existing sliding-window algorithm for (unconstrained) diversity maximization [14]. In Section 5.2, we propose our extended sliding-window algorithms for FDM based on the algorithms in Section 4 and Section 5.1.

### 5.1. (Unconstrained) Sliding-Window Algorithm

The unconstrained sliding-window algorithm is shown in Algorithm 5 and illustrated in Figure 5. First of all, it keeps two sequences Λ,U, both ranging from $d_{min}$ to $d_{max}$, to guess the optimum OPT[W] of DM on the window *W* (Line 1). For each combination of λ∈Λ and μ∈U, it initializes two candidate solutions $A_{\lambda ,\mu }$ and $B_{\lambda ,\mu }$, each of which will be maintained by Algorithm 1 on two consecutive sub-sequences of *X*. Two sets $A_{\lambda ,\mu }^{\prime }$ and $B_{\lambda ,\mu }^{\prime }$ to store the replacements of the elements in $A_{\lambda ,\mu }$ and $B_{\lambda ,\mu }$, in case that they fall out of the sliding window, are also initialized as empty sets (Lines 2 and 3). Then, for each element x∈X, it adds *x* to each $B_{\lambda ,\mu }$ using the same method as Algorithm 1. Once *x* is added to $B_{\lambda ,\mu }$, it will be set as its own replacement in $B_{\lambda ,\mu }^{\prime }$ (Lines 7 and 8). Otherwise, it checks whether the distance between *x* and any existing element in $B_{\lambda ,\mu }$ is, at most, μ and assigns *x* as the replacement of such an element in $B_{\lambda ,\mu }^{\prime }$ (Line 10). Similarly, it also checks whether *x* can replace any element in $A_{\lambda ,\mu }$ and perform the assignment if so (Line 12). After that, if the diversity of any candidate $B_{\lambda ,\mu }$ with $|B_{\lambda ,\mu }|=k$ exceeds λ, it will remove *x* from $B_{\lambda ,\mu },B_{\lambda ,\mu }^{\prime }$ and set them as $A_{\lambda ,\mu },A_{\lambda ,\mu }^{\prime }$, and then re-initialize a new $B_{\lambda ,\mu }$ and $B_{\lambda ,\mu }^{\prime }$ with *x* (Lines 13–16). We describe the post-processing procedure for the window *W* containing the last *w* elements in *X*, which can be easily extended to any window $W^{\left(T\right)}$ at time *T*, in Lines 17–23. It considers two cases for different values of λ,μ: *(i)* when $A_{\lambda ,\mu }\subseteq W$, it runs any algorithm ALG for (centralized) max–min diversity maximization on $A_{\lambda ,\mu }\cup B_{\lambda ,\mu }$ to find a size-*k* candidate solution $S_{\lambda ,\mu }$ (Line 20); *(ii)* when $B_{\lambda ,\mu }\subseteq W$, ALG is run on $\left(W\cap A_{\lambda ,\mu }^{\prime }\right)\cup B_{\lambda ,\mu }$, i.e., the non-expired elements from $A_{\lambda ,\mu }^{\prime }$ and $B_{\lambda ,\mu }$, instead (Line 22). Finally, the best solution found after post-processing all candidates is returned as the solution *S* for the window *W* (Line 23).
**Algorithm 5** SWDM**Input**:Stream *X*, distance metric d(·,·), window size $w\in Z^{+}$, parameter ε∈(0,1), solution size $k\in Z^{+}$**Output**:A set S⊆W with |S|=k1:$\Lambda ,U=\{\frac{d_{min}}{{\left(1-\varepsilon \right)}^{j}}\;:\;j\in Z_{0}^{+}\wedge {\left(1-\varepsilon \right)}^{j}\geq \frac{d_{min}}{d_{max}}\}$2:**for all**λ∈Λ and μ∈U **do**3:    Initialize $A_{\lambda ,\mu },A_{\lambda ,\mu }^{\prime }=\emptyset$ and $B_{\lambda ,\mu },B_{\lambda ,\mu }^{\prime }=\emptyset$4:**for all** x∈X **do**5:    **for all** λ∈Λ **do**6:        **for all** μ∈U **do**7:           **if** $|B_{\lambda ,\mu }|<k$ and $d(x,B_{\lambda ,\mu })\geq \mu$ **then**8:               $B_{\lambda ,\mu }\leftarrow B_{\lambda ,\mu }\cup \left\{x\right\}$, $B_{\lambda ,\mu }^{\prime }\left[x\right]\leftarrow x$9:           **else if** $d(x,B_{\lambda ,\mu })<\mu$ **then**10:               $y^{\prime }\leftarrow {arg min}_{y\in B_{\lambda ,\mu }}d\left(x,y\right)$, $B_{\lambda ,\mu }^{\prime }\left[y^{\prime }\right]\leftarrow x$11:           **if** $A_{\lambda ,\mu }\neq \emptyset$ and $d(x,A_{\lambda ,\mu })<\mu$ **then**12:               $y^{\prime }\leftarrow {arg min}_{y\in A_{\lambda ,\mu }}d\left(x,y\right)$, $A_{\lambda ,\mu }^{\prime }\left[y^{\prime }\right]\leftarrow x$13:        **if** $\max_{\mu \in U:|B_{\lambda ,\mu }|=k}div\left(B_{\lambda ,\mu }\right)>\lambda$ **then**14:           Remove *x* from each $B_{\lambda ,\mu },B_{\lambda ,\mu }^{\prime }$15:           $A_{\lambda ,\mu },A_{\lambda ,\mu }^{\prime }\leftarrow B_{\lambda ,\mu },B_{\lambda ,\mu }^{\prime }$ for each μ∈U16:           $B_{\lambda ,\mu },B_{\lambda ,\mu }^{\prime }\leftarrow \left\{x\right\}$ for each μ∈U▹Post-processing17:W←{x∈X:max{1,|X|−w+1}≤t(x)≤|X|}18:**for all**λ∈Λ and μ∈U **do**19:    **if** $A_{\lambda ,\mu }\subseteq W$ **then**20:        $S_{\lambda ,\mu }\leftarrow \mathrm{ALG}\left(k,A_{\lambda ,\mu }\cup B_{\lambda ,\mu }\right)$21:    **else if** $B_{\lambda ,\mu }\subseteq W$ **then**22:        $S_{\lambda ,\mu }\leftarrow \mathrm{ALG}\left(k,\left(W\cap A_{\lambda ,\mu }^{\prime }\right)\cup B_{\lambda ,\mu }\right)$23:**return** $S\leftarrow {arg max}_{\lambda \in \Lambda ,\mu \in U:|S_{\lambda ,\mu }|=k}div\left(S_{\lambda ,\mu }\right)$

### 5.2. Fair Sliding-Window Algorithms

Generally, to extend SFDM1 and SFDM2 so that they can work in the sliding-window model, we need to modify them in two aspects: *(i)* the stream processing should follow the procedure of Algorithm 5 instead of Algorithm 1 to maintain the candidate solutions for the case when old elements are deleted from the window *W*; *(ii)* the post-processing should be adjusted for the candidate solutions kept by Algorithm 5 during stream processing with theoretical guarantees.

Specifically, the procedures of our extended algorithms, i.e., SWFDM1 and SWFDM2 are presented in Algorithm 6. Here, we put the descriptions of both algorithms together because they share many common subroutines and inherit some others from Algorithms 2–5. Following the procedure of Algorithm 5, they initialize the candidate solutions for different guesses λ,μ of OPT[W] in the sequences Λ and U. In the stream processing (Lines 1–11), SWFDM1 and SWFDM2 adopts the same method as used in Algorithm 5 to maintain the unconstrained candidate solutions as well as the monochromatic candidate solutions for each group i∈[m]. The only difference is the solution size of each monochromatic candidate, which is $k_{i}$ for i∈{1,2} in SWFDM1 but *k* for each i∈[m] in SWFDM2.

The following theorem indicates the approximation factor of Algorithm 5.

**Theorem** **6**.
*Algorithm 5 is a $\frac{\xi -\varepsilon }{5}$-approximation algorithm for max–min diversity maximization when a ξ-approximation algorithm ALG for (centralized) max–min diversity maximization is used for post-processing.*


We refer readers to Lemma 4.7 in [14] for the proof of Theorem 6. Here, if GMM [39], which is $\frac{1}{2}$-approximate for max–min diversity maximization, is used as ALG, the approximation factor of Algorithm 5 will be $\frac{1-\varepsilon }{10}$. In terms of complexity, Algorithm 5 stores $O(\frac{k\log^{2}\Delta }{\varepsilon^{2}})$ elements, takes $O(\frac{k\log^{2}\Delta }{\varepsilon^{2}})$ time per element for stream processing, and spends $O(\frac{k^{2}\log^{2}\Delta }{\varepsilon^{2}})$ time for post-processing.
**Algorithm 6** SWFDM**Input**:Stream $X={⋃}_{i=1}^{m}X_{i}$, distance metric d(·,·), parameter ε∈(0,1), window size $w\in Z^{+}$, size constraints $k_{1},\ldots ,k_{m}$ ($k=\sum_{i=1}^{m}k_{i}$)**Output**:A set S⊆W s.t. $\left|S\cap X_{i}\right|=k_{i}$ for i∈[m]▹Stream processing1:$\Lambda ,U=\{\frac{d_{min}}{{\left(1-\varepsilon \right)}^{j}}\;:\;j\in Z_{0}^{+}\wedge {\left(1-\varepsilon \right)}^{j}\geq \frac{d_{min}}{d_{max}}\}$2:**for all** λ∈Λ,μ∈U **do**3:    Initialize $A_{\lambda ,\mu },A_{\lambda ,\mu }^{\prime },B_{\lambda ,\mu },B_{\lambda ,\mu }^{\prime }=\emptyset$4:    **for all** i∈[m] **do**5:        Initialize $A_{\lambda ,\mu }^{\left(i\right)},A_{\lambda ,\mu }^{\prime (i)},B_{\lambda ,\mu }^{\left(i\right)},B_{\lambda ,\mu }^{\prime (i)}=\emptyset$6:**for all** x∈X **do**7:    Run Lines 5–16 of Algorithm 5 to update $A_{\lambda ,\mu }$, $A_{\lambda ,\mu }^{\prime }$, $B_{\lambda ,\mu }$, and $B_{\lambda ,\mu }^{\prime }$ w.r.t. *x*8:    **if** m=2∧c(x)=i and ‘SWFDM1’ is used **then**9:        Run Lines 5–16 of Algorithm 5 to update to update $A_{\lambda ,\mu }^{\left(i\right)}$, $A_{\lambda ,\mu }^{\prime (i)}$, $B_{\lambda ,\mu }^{\left(i\right)}$, and $B_{\lambda ,\mu }^{\prime (i)}$ w.r.t. *x* under size constraint $k_{i}$10:    **else if** c(x)=i and ‘SWFDM2’ is used **then**11:        Run Lines 5–16 of Algorithm 5 to update to update $A_{\lambda ,\mu }^{\left(i\right)}$, $A_{\lambda ,\mu }^{\prime (i)}$, $B_{\lambda ,\mu }^{\left(i\right)}$, and $B_{\lambda ,\mu }^{\prime (i)}$ w.r.t. *x* under size constraint *k*▹Post-processing12:W←{x∈X:max{1,|X|−w+1}≤t(x)≤|X|}13:**for all**λ∈Λ and μ∈U **do**14:    **if** $A_{\lambda ,\mu }\subseteq W$ **then**15:        $S_{\lambda ,\mu }\leftarrow \mathrm{ALG}\left(k,A_{\lambda ,\mu }\cup B_{\lambda ,\mu }\right)$16:    **else if** $B_{\lambda ,\mu }\subseteq W$ **then**17:        $S_{\lambda ,\mu }\leftarrow \mathrm{ALG}\left(k,\left(W\cap A_{\lambda ,\mu }^{\prime }\right)\cup B_{\lambda ,\mu }\right)$18:    **if** m=2 and ‘SWFDM1’ is used **then**19:        **if** $\left|S_{\lambda ,\mu }\right|=k\wedge \left|S_{\lambda ,\mu }\cap X_{i}\right|<k_{i}$ **then**20:           **if** $A_{\lambda ,\mu }^{\left(i\right)}\subseteq W$ **then**21:               $S_{\lambda ,\mu }^{\left(i\right)}\leftarrow \mathrm{ALG}\left(k_{i},A_{\lambda ,\mu }^{\left(i\right)}\cup B_{\lambda ,\mu }^{\left(i\right)}\right)$22:           **else if** $B_{\lambda ,\mu }^{\left(i\right)}\subseteq W$ **then**23:               $S_{\lambda ,\mu }^{\left(i\right)}\leftarrow \mathrm{ALG}\left(k_{i},\left(W\cap A_{\lambda ,\mu }^{\prime (i)}\right)\cup B_{\lambda ,\mu }^{\left(i\right)}\right)$24:           Run Lines 10–15 of Algorithm 2 using $S_{\lambda ,\mu }$ and $S_{\lambda ,\mu }^{\left(i\right)}$ as input to find a fair solution $S_{\lambda ,\mu }$25:    **else if** ‘SWFDM2’ is used **then**26:        **for** i∈[m] **do**17:           **if** $A_{\lambda ,\mu }^{\left(i\right)}\subseteq W$ **then**28:               $S_{\lambda ,\mu }^{\left(i\right)}\leftarrow \mathrm{ALG}\left(k,A_{\lambda ,\mu }^{\left(i\right)}\cup B_{\lambda ,\mu }^{\left(i\right)}\right)$29:           **else if** $B_{\lambda ,\mu }^{\left(i\right)}\subseteq W$ **then**30:               $S_{\lambda ,\mu }^{\left(i\right)}\leftarrow \mathrm{ALG}\left(k,\left(W\cap A_{\lambda ,\mu }^{\prime (i)}\right)\cup B_{\lambda ,\mu }^{\left(i\right)}\right)$31:        Run Lines 10–17 (with $d\left(x,y\right)<\frac{\xi \mu }{m+1}$ in Line 13) of Algorithm 3 using $S_{\lambda ,\mu }$ and $S_{all}=$${⋃}_{i=1}^{m}S_{\lambda ,\mu }^{\left(i\right)}\cup S_{\lambda ,\mu }$ as input to find a fair solution $S_{\lambda ,\mu }$32:**return** $S\leftarrow {arg max}_{\lambda \in \Lambda ,\mu \in U:|S_{\lambda ,\mu }|=k}div\left(S_{\lambda ,\mu }\right)$

The post-processing steps of both algorithms for the window *W* containing the last *w* elements in *X* are shown in Lines 12–31. Note that these steps can be trivially used for any window $W^{\left(T\right)}$ based on the intermediate candidate solutions at time *T*. It first computes an unconstrained solution $S_{\lambda ,\mu }$ for each λ∈Λ and μ∈U from the (unconstrained) candidates kept during stream processing based on Algorithm 5. For SWFDM1, it next checks whether $S_{\lambda ,\mu }$ has contained *k* elements and an under-filled group exists in $S_{\lambda ,\mu }$. If $|S_{\lambda ,\mu }|<k$, the post-processing procedure is skipped because $S_{\lambda ,\mu }$ cannot produce any valid solution. Moreover, if $|S_{\lambda ,\mu }|=k$ and has already satisfied the fairness constraint, the post-processing will not be required anymore. Otherwise, it computes a group-specific solution $S_{\lambda ,\mu }^{\left(i_{u}\right)}$ of size $k_{i_{u}}$ from the candidates maintained for the under-filled group $i_{u}$ and performs the procedure as Lines 10–15 of Algorithm 2 to greedily swap the elements from $S_{\lambda ,\mu }^{\left(i_{u}\right)}$ into $S_{\lambda ,\mu }$ and the elements from the over-filled group $i_{o}$ out of $S_{\lambda ,\mu }$ so that $S_{\lambda ,\mu }$ becomes a fair solution. For SWFDM2, it computes a group-specific solution $S_{\lambda ,\mu }^{\left(i\right)}$ of size *k* from each group-specific candidate for i∈[m]. $S_{\lambda ,\mu }$ as well as each $S_{\lambda ,\mu }^{\left(i\right)}$ constitutes $S_{all}$ for post-processing. Then, using the same method as Algorithm 3, it picks a subset $S_{\lambda ,\mu }^{\prime }$ of $S_{\lambda ,\mu }$, divides $S_{all}$ into clusters, and augments $S_{\lambda ,\mu }^{\prime }$ via matroid intersection as the new solution $S_{\lambda ,\mu }$. Both algorithms return the fair solution with maximum diversity after post-processing as the final solution for FDM on the window *W*.

**Theoretical Analysis:** Subsequently, we will analyze the theoretical soundness and complexities of the extended SWFDM1 and SWFDM2 algorithms for FDM in the sliding-window model by generalizing the analyses for SFDM1 and SFDM2 in Section 4.

**Theorem** **7**.
SWFDM1
*is a $\frac{\xi -\varepsilon }{10}$-approximation algorithm for FDM in the sliding-window model when a ξ-approximation algorithm is used for post-processing. It keeps $O(\frac{k\log^{2}\Delta }{\varepsilon^{2}})$ elements, takes $O(\frac{k\log^{2}\Delta }{\varepsilon^{2}})$ time per element in streaming processing and $O(\frac{k^{2}\log^{2}\Delta }{\varepsilon^{2}})$ time for post-processing.*


**Proof.** First, based on the analyses in [14], when $\mu \leq \frac{\mathrm{OPT}[W]}{5}$, there exists $\lambda^{\prime }\in \Lambda$ such that $div(S_{\lambda^{\prime },\mu })\geq \xi \mu$. Let $\mu^{\prime }$ be the value in U such that $\mu^{\prime }\in \left[\left(1-\varepsilon \right)\cdot \frac{{\mathrm{OPT}}_{f}\left[W\right]}{5},\frac{{\mathrm{OPT}}_{f}\left[W\right]}{5}\right]$, where ${\mathrm{OPT}}_{f}\left[W\right]$ is the optimal diversity for FDM on window *W*. Obviously, ${\mathrm{OPT}}_{f}\left[W\right]\leq \mathrm{OPT}\left[W\right]$. Accordingly, we can find the values $\lambda^{\prime }\in \Lambda$ and $\mu^{\prime }\in U$ with $div\left(S_{\lambda^{\prime },\mu^{\prime }}\right)\geq \xi \mu^{\prime }$. Then, Lemma 2 guarantees that $div\left(S_{\lambda^{\prime },\mu^{\prime }}\right)\geq \frac{\xi \mu^{\prime }}{2}$ after the post-processing procedure. Combining the above results, we have $div\left(S\right)\geq div\left(S_{\lambda^{\prime },\mu^{\prime }}\right)\geq \frac{(1-\varepsilon )\xi }{10}\cdot {\mathrm{OPT}}_{f}\left[W\right]$, where *S* is the solution for FDM on *W* returned by SWFDM1. Finally, since the number of candidates increases from $O(\frac{\log\Delta }{\varepsilon })$ to $O(\frac{\log^{2}\Delta }{\varepsilon^{2}})$ and the complexities of the remaining steps are not changed, the time and space complexities of SWFDM1 grow by a factor of $\frac{\log\Delta }{\varepsilon }$ compared with SFDM1. □

**Theorem** **8**.
SWFDM2
*is a $\frac{\xi -\varepsilon }{15m+10}$-approximation algorithm for FDM in the sliding-window model when a ξ-approximation algorithm is used for post-processing. It keeps $O(\frac{km\log^{2}\Delta }{\varepsilon^{2}})$ elements in memory, and takes $O(\frac{k\log^{2}\Delta }{\varepsilon^{2}})$ time per element in streaming processing and $O\left(\frac{k^{2}m\log^{2}\Delta }{\varepsilon^{2}}\cdot \left(m+\log^{2}k\right)\right)$ time for post-processing.*


**Proof.** Similar to the proof of Theorem 7, we find the values $\lambda^{\prime }\in \Lambda$ and $\mu^{\prime }\in U$ such that $\mu^{\prime }\in \left[\left(1-\varepsilon \right)\cdot \frac{{\mathrm{OPT}}_{f}\left[W\right]}{5},\frac{{\mathrm{OPT}}_{f}\left[W\right]}{5}\right]$ and $div\left(S_{\lambda^{\prime },\mu^{\prime }}\right)\geq \xi \mu^{\prime }$, where ${\mathrm{OPT}}_{f}\left[W\right]$ is the optimal diversity value for FDM on *W*. Then, Lemmas 3 and 4 guarantee that $div\left(S_{\lambda^{\prime },\mu^{\prime }}\right)\geq \frac{\xi \mu^{\prime }}{3m+2}$ after the post-processing procedure. Combining the above results, we have $div\left(S\right)\geq div\left(S_{\lambda^{\prime },\mu^{\prime }}\right)\geq \frac{(1-\varepsilon )\xi }{15m+10}\cdot {\mathrm{OPT}}_{f}\left[W\right]$, where *S* is the solution for FDM on *W* returned by SWFDM2. Since the number of candidates increases from $O(\frac{\log\Delta }{\varepsilon })$ to $O(\frac{\log^{2}\Delta }{\varepsilon^{2}})$ and the complexities of the remaining steps are not changed, the time and space complexities of SWFDM2 grows by a factor of $\frac{\log\Delta }{\varepsilon }$ compared with SFDM2. □

Finally, since the approximation factor ξ of the algorithm ALG we use is Θ(1), e.g., $\xi =\frac{1}{2}$ for GMM [39], the approximation factors of SWFDM1 and SWFDM2 are written as Θ(1) and $\Theta (m^{-1})$, respectively, for simplicity.

## 6. Experiments

In this section, we evaluate the performance of our proposed algorithms on several real-world and synthetic datasets. We first introduce our experimental setup in Section 6.1. Then, experimental results in the streaming setting are presented in Section 6.2. Finally, experimental results in the sliding-window setting are presented in Section 6.3.

### 6.1. Experimental Setup


**Datasets:** Our experiments are conducted on four publicly available real-world datasets, as follows:



**Adult** (https://archive.ics.uci.edu/dataset/2/adult, accessed on 12 July 2023) is a collection of 48,842 records from the 1994 US Census database. We select six numeric attributes as features and normalize each of them to have zero mean and unit standard deviation. The Euclidean distance is used as the distance metric. The groups are generated from two demographic attributes: sex and race. By using them individually and in combination, there are two (sex), five (race), and ten (sex + race) groups, respectively.**CelebA** (https://mmlab.ie.cuhk.edu.hk/projects/CelebA.html, accessed on 12 July 2023) is a set of 202,599 images of human faces. We use 41 pre-trained class labels as features and the Manhattan distance as the distance metric. We generate two groups from sex {‘female’, ‘male’}, two groups from age {‘young’, ‘not young’}, and four groups from their combination, respectively.**Census** (https://archive.ics.uci.edu/dataset/116/us+census+data+1990, accessed on 12 July 2023) is a set of 2,426,116 records from the 1990 US Census data. We take 25 (normalized) numeric attributes as features and use the Manhattan distance as the distance metric. We generate 2, 7, and 14 groups from sex, age, and both of them, respectively.**Lyrics** (http://millionsongdataset.com/musixmatch, accessed on 12 July 2023) is a set of 122,448 documents, each of which is the lyrics of a song. We train a topic model with 50 topics using LDA [45] implemented in Gensim (https://radimrehurek.com/gensim, accessed on 12 July 2023). Each document is represented as a 50-dimensional vector and the angular distance is used as the distance metric. We generate 15 groups based on the primary genres of songs.


We also generate different synthetic datasets with varying *n* and *m* for scalability tests. In each synthetic dataset, we generate ten two-dimensional Gaussian isotropic blobs with random centers in ${\left[-10,10\right]}^{2}$ and identity covariance matrices. We assign points to groups uniformly at random. The Euclidean distance is used as the distance metric. The number *n* of points varies from $10^{3}$ to $10^{7}$ with fixed m=2 or 10. The number *m* of groups varies from 2 to 20 with fixed $n=10^{5}$. The statistics of all datasets are summarized in Table 1.

**Algorithms:** We compare our streaming algorithms, i.e., SFDM1 and SFDM2, and sliding-window algorithms, i.e., SWFDM1 and SWFDM2, with four existing offline FDM algorithms: the $\frac{1}{3m-1}$-approximation FairFlow algorithm for an arbitrary *m*, the $\frac{1}{5}$-approximation FairGMM algorithm for small *k* and *m*, and the $\frac{1}{4}$-approximation FairSwap algorithm specific for m=2 in [17], and the $\frac{1-\varepsilon }{m+1}$-approximation FairGreedyFlow algorithm for an arbitrary *m* in [20]. Since no implementation for the algorithms in [17,20] is available, we implement them by ourselves, following the description in the original paper. All the algorithms are implemented in Python 3. All experiments were run on a desktop with an Intel^®^ Core^™^ i5-9500 3.0GHz processor and 32GB RAM running Ubuntu 20.04.3 LTS. Each algorithm was run on a single thread.

For a given solution size *k*, the group-specific size constraint $k_{i}$ for each group i∈[m] is set based on equal representation, which has been widely used in the literature [21,22,23,27]: If *k* is divisible by *m*, $k_{i}=\frac{k}{m}$ for each i∈[m]. If *k* is not divisible by *m*, $k_{i}=\left\lceil \frac{k}{m}\right\rceil$ for some groups or $\left\lfloor \frac{k}{m}\right\rfloor$ for the others while ensuring $\sum_{i=1}^{m}k_{i}=k$. We also compare the performance of different algorithms for proportional representation [8,26,27], another popular notion of fairness that requires a proportion of elements from each group in the solution that are generally preserved in the dataset.

**Performance Metrics:** The performance of each algorithm is evaluated in terms of efficiency, quality, and space usage. The efficiency is measured as average update time, i.e., the average wall-clock time used to compute a solution for each arrival element in the stream. The quality is measured by the value of the diversity function of the solution returned by an algorithm. Since computing the optimal diversity ${\mathrm{OPT}}_{f}$ of FDM is infeasible, we run GMM [39] for unconstrained diversity maximization to estimate an upper bound of ${\mathrm{OPT}}_{f}$ for comparison. Space usage is measured by the number of distinct elements stored by each algorithm. However, only the numbers of elements stored by our proposed algorithms are presented because offline algorithms should keep all elements in memory for random access, and thus their space usage is always equal to the dataset (or window) size. We run each experiment 10 times with different permutations of the same dataset and report the average of each measure over 10 runs.

### 6.2. Results in Streaming Setting


**Effect of Parameter ε:**Figure 6 illustrates the performance of SFDM1 and SFDM2 with different values of ε when k=20. We range the value of ε from 0.05 to 0.25 on *Adult*, *CelebA*, and *Census* and from 0.02 to 0.1 on *Lyrics*. Since the angular distances between any two vectors are at most $\frac{\pi }{2}$, larger values of ε (e.g., >0.1) leads to greater estimation errors for ${\mathrm{OPT}}_{f}$ and thus significantly lower solution quality in *Lyrics*. Generally, SFDM1 has higher efficiency and smaller space usage than SFDM2 for different values of ε, but SFDM2 exhibits a better solution quality. Furthermore, the running time and numbers of stored elements of both algorithms significantly decrease when the value of ε increases. This is consistent with our analyses in Section 4 because the number of guesses for ${\mathrm{OPT}}_{f}$, and thus the number of candidates maintained by both algorithms is $O(\frac{\log\Delta }{\varepsilon })$. A slightly surprising result is that the diversity values of the solutions do not obviously degrade, even when ε=0.25. This can be explained by the fact that both algorithms return the best solutions after post-processing among all candidates, which means that they provide good solutions as long as there is some μ∈U close to ${\mathrm{OPT}}_{f}$. We infer that μ still exists when ε=0.25. Nevertheless, we note that the chance of finding an appropriate value of μ will be smaller when the value of ε is larger, which will result in a less stable solution quality. Therefore, in the experiments for streaming setting, we always use ε=0.1 for both algorithms on all datasets, except *Lyrics*, where the value of ε is set to 0.05. The impact of ε on the performance of SWFDM1 and SWFDM2 is generally similar to that of SFDM1 and SFDM2. However, since the number of candidate solutions is quadratic with respect to ε, we use a larger ε=0.25 for SWFDM1 and SWFDM2 on all datasets except *Lyrics*, where the value of ε is set to 0.1.



**Overview:**Table 2 presents the performance of different algorithms for FDM in the streaming setting on four real-world datasets with different group partitions when the solution size *k* is fixed to 20. FairGMM is not included because it needs to enumerate, at most, kmk=O(em)k candidates for solution computation and cannot scale to k>10 and m>5. First, compared with the unconstrained solution returned by GMM, all the fair solutions are less diverse because of the additional fairness constraints. Since GMM is a $\frac{1}{2}$-approximation algorithm and $\mathrm{OPT}\geq {\mathrm{OPT}}_{f}$, 2·div_GMM is the upper bound of ${\mathrm{OPT}}_{f}$, from which we observe that all five fair algorithms return solutions with much better approximation ratios than their lower bounds. In case of m=2, SFDM1 runs the fastest among all five algorithms, which achieves a speed-up of from two to four orders of magnitude over FairSwap, FairFlow and FairGreedyFlow. At the same time, its solution quality is close to or equal to that of FairSwap in most cases. SFDM2 shows lower efficiency than SFDM1 due to the higher cost of post-processing. However, it is still much more efficient than offline algorithms by taking advantage of stream processing. In addition, the solution quality of SFDM2 benefits from the greedy selection procedure in Algorithm 4, which is not only consistently better than that of SFDM1 but also better than that of FairSwap on the *Adult* and *Census* datasets. In the case of m>2, SFDM1 and FairSwap are not applicable anymore. In addition, FairGreedyFlow cannot finish within one day on the *Census* dataset, and the corresponding results are also ignored. SFDM2 shows significant advantages compared to FairFlow and FairGreedyFlow in terms of both solution quality and efficiency. It provides up to 3.4 times more diverse solutions than FairFlow and FairGreedyFlow while running several orders of magnitude faster. In terms of space usage, both SFDM1 and SFDM2 store very small portions of elements (<0.1% on *Census*) on all datasets. SFDM2 contains slightly more elements than SFDM1 because the capacity of each group-specific candidate for group *i* is set to *k* instead of $k_{i}$. For SFDM2, the number of stored elements increases nearly linearly with *m* since the number of candidates is linear to *m*.



**Effect of Solution Size *k*:** The impact of solution size *k* on the performance of different algorithms in the streaming setting is shown in Figure 7 and Figure 8. Here, we vary *k* in [5,50] when m≤5, or [10,50] when 5<m≤10, or [15,50] when m>10, since we restrict that an algorithm must pick at least one element from each group. For each algorithm, the diversity value drops with *k* as the diversity function is monotonically non-increasing. At the same time, the update time grows with *k*, as their time complexities are linear or quadratic w.r.t. *k*. Compared with the solutions of GMM, all fair solutions are slightly less diverse when m=2. The gaps in diversity values become more apparent when *m* is larger. Although FairGMM achieves a slightly higher solution quality than any other algorithm when k≤10 and m=2, it is not scalable to a larger *k* and *m* due to the enormous cost of enumeration. The solution quality of FairSwap, SFDM1, and SFDM2 is close to each other when m=2, which is better than that of FairFlow and FairGreedyFlow. However, the efficiencies of SFDM1 and SFDM2 are orders of magnitude higher than those of offline algorithms. Furthermore, when m>2, SFDM2 outperforms FairFlow and FairGreedyFlow in terms of both efficiency and effectiveness across all *k* values. However, since the time complexity of SFDM2 is quadratic w.r.t. both *k* and *m*, its update time increases drastically with *k* and might be close to that of FairFlow when *k* and *m* are large.



**Scalability:** We evaluate the scalability of each algorithm in the streaming setting on synthetic datasets by varying the dataset size *n* from $10^{3}$ to $10^{7}$ and the number of groups *m* from 2 to 20. The results regarding solution quality and update time for different values of *n* and *m* when k=20 are presented in Figure 9. First of all, SFDM2 shows a much better scalability than FairFlow and FairGreedyFlow w.r.t. *m* in terms of solution quality. The diversity value of the solution of SFDM2 only slightly decreases with *m* and is up to 3 times higher than that of FairFlow and FairGreedyFlow when m>10. However, its update time increases more rapidly with *m* due to the quadratic dependence on *m*. Moreover, the diversity values of different algorithms slightly grow with *n* but are always close to each other for different values of *n* when m=2. Finally, the running time of offline algorithms is linear to *n*. However, the update time of SFDM1 and SFDM2 are almost independent of *n*, as analyzed in Section 4.



**Equal vs. Proportional Representation:**Figure 10 compares the solution quality and running time of different algorithms for two popular notions of fairness, i.e., equal representation (ER) and proportional representation (PR), when k=20 on *Adult* with highly skewed groups, where 67% of the records are for males and 87% of the records are for Whites. The diversity value of the solution of each algorithm is slightly higher for PR than ER, as the solution for PR is closer to the unconstrained one. The running time of SFDM1 and SFDM2 is slightly shorter for PR than ER, since fewer swapping and augmentation steps are performed on each candidate during post-processing. The results for SWFDM1 and SWFDM2 are similar and will be omitted.


### 6.3. Results in Sliding-Window Setting


**Overview:**Table 3 shows the performance of different algorithms for sliding-window FDM on four real-world datasets with different group settings when the solution size *k* is fixed to 20 and the window size *w* is set to 25 k on *Adult* (as its size is smaller than 100 k) or 100 k on other datasets. FairGMM is also omitted in Table 3 due to its high complexity. Compared with the streaming setting, the “price of fairness” becomes higher in the sliding-window setting, for two possible reasons. First, the approximation factors of our proposed algorithms are lower. Second, some minor groups contain too few elements in the window when the value of *m* is large (marginally larger than $k_{i}$). Thus, the selection of elements from such groups is very restricted to ensure fairness. Nevertheless, we still find that all fair algorithms provide solutions with much better approximations than their lower bounds.


We observe that SWFDM2 runs the fastest of all five algorithms, which achieves 5–150× speedups over FairSwap, FairFlow and FairGreedyFlow. Moreover, SWFDM1 and SWFDM2 have a slightly lower solution quality than FairSwap when m=2. Nevertheless, SWFDM2 shows significant advantages over FairFlow and FairGreedyFlow in terms of both solution quality and efficiency when m>2. Unlike the streaming setting, SWFDM2 shows higher efficiency than SWFDM1. This is because SWFDM2 maintains group-specific solutions with size constraints *k*, instead of $k_{i}$ for SWFDM1, in stream processing. Consequently, its group-specific solutions often expire (i.e., $A_{\lambda ,\mu }^{\left(i\right)}⊈W$), and thus are not eligible for post-processing. However, such efficiency improvements come at the expense of less diverse solutions. In terms of space usage, both SWFDM1 and SWFDM2 store very small portions of elements (at most 3.2%·w) across all datasets. SWFDM2 keeps slightly more elements than SWFDM1 also because the capacity of each group-specific solution is *k* instead of $k_{i}$.

**Effect of Solution Size *k*:** The impact of solution size *k* on the performance of different algorithms in the sliding-window setting is illustrated in Figure 11 and Figure 12. We use the same values of *k* as in the streaming setting. The window size *w* is set to w=25 k for *Adult* and 100 k for others. For each algorithm, the diversity value drops with *k* as the diversity function is monotonically non-increasing. At the same time, the update time grows with *k* as their time complexities are linear or quadratic w.r.t. *k*. The gaps in diversity values between unconstrained and fair solutions are much larger than those in the streaming setting. The reasons for this were explained in the previous paragraph. The solution quality of SWFDM1 and SWFDM2 is slightly lower than FairSwap when m=2, but is still better than that of FairFlow and FairGreedyFlow. However, their efficiencies are always much higher than those of offline algorithms. Finally, when m>2, SWFDM2 outperforms FairFlow and FairGreedyFlow in terms of efficiency and effectiveness across all *k* values.

**Scalability:** We evaluate the scalability of each algorithm in the sliding-window setting on synthetic datasets by varying the number of groups *m* from 2 to 20 and the window size *w* from $10^{3}$ to $10^{6}$. The results regarding solution quality and update time for different values of *w* and *m* when k=20 are presented in Figure 13. First of all, SWFDM2 shows a much better scalability than FairFlow and FairGreedyFlow w.r.t. *m* in terms of solution quality. The diversity value of the solution of SWFDM2 only slightly decreases with *m*. However, for FairFlow and FairGreedyFlow, the diversity values drop drastically with *m*. Nevertheless, its update time increases more rapidly with *m* since its time complexity is quadratic w.r.t. *m*. Furthermore, the results of the diversity values of different algorithms with varying *w* are similar to those for varying *k*. As expected, the running time of offline algorithms is nearly linear to *w*. However, unlike the streaming setting, the update time of SWFDM1 and SWFDM2 increases with *w* because more candidates are non-expired and thus considered in post-processing for a larger value of *w*.

## 7. Conclusions

In this paper, we studied the diversity maximization problem with fairness constraints in the streaming and sliding window settings. We first proposed a $\frac{1-\varepsilon }{4}$-approximation streaming algorithm for this problem when there were two groups in the dataset and a $\frac{1-\varepsilon }{3m+2}$-approximation streaming algorithm that could deal with an arbitrary number *m* of groups. Moreover, we extended the two proposed streaming algorithms to the sliding-window model while maintaining approximation factors of Θ(1) and $\Theta (m^{-1})$, respectively. Extensive experiments on real-world and synthetic datasets confirmed the efficiency, effectiveness, and scalability of our proposed algorithms.

In future work, we would like to improve the approximation ratios of the proposed algorithms. It would also be interesting to consider diversity maximization problems with other objective functions and fairness constraints defined on multiple sensitive attributes.

## Figures and Tables

**Figure 1 entropy-25-01066-f001:**
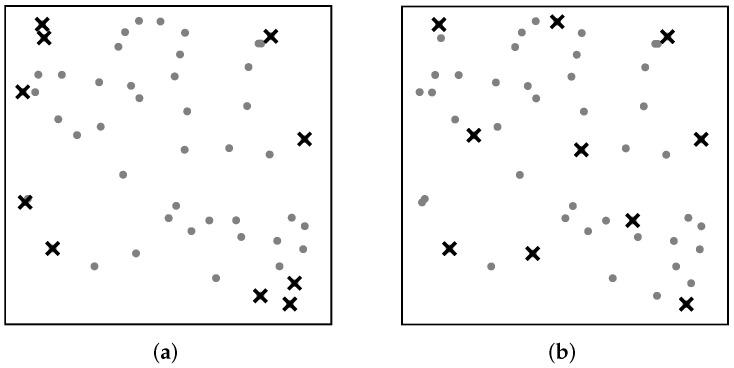
Comparison of (**a**) max–sum dispersion (MSD) and (**b**) max–min dispersion (MMD) for diversity maximization on a dataset of one hundred points. We use circles and crossmarks to denote all points in the dataset and the points selected based on MSD and MMD.

**Figure 2 entropy-25-01066-f002:**
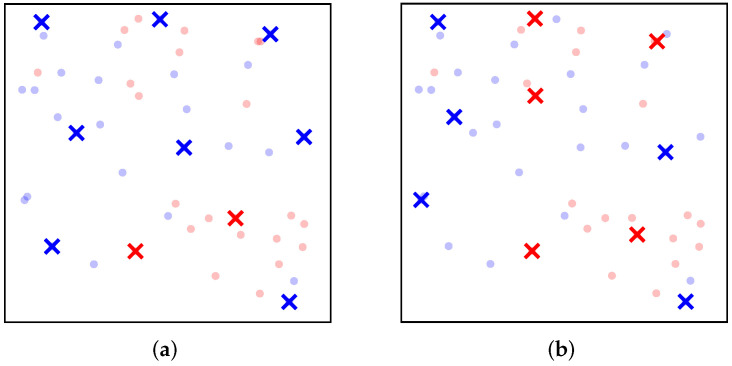
Comparison of (**a**) unconstrained max–min diversity maximization and (**b**) fair max–min diversity maximization. We have a set of individuals, each described by two attributes, partitioned into two disjoint groups of red and blue, respectively. Fair diversity maximization returns a subset of size 10 that maximizes diversity in terms of attributes and contains an equal number (i.e., $k_{i}=5$) of elements from both groups.

**Figure 3 entropy-25-01066-f003:**
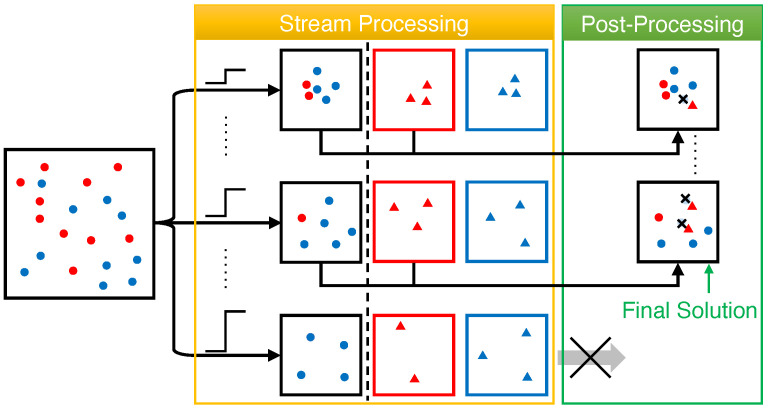
Illustration of the SFDM1 algorithm. During stream processing, one group-blind and two group-specific candidates are maintained for each guess μ of ${\mathrm{OPT}}_{f}$. Then, a subset of group-blind candidates is selected for post-processing by adding the elements from the under-filled group before deleting the elements from the over-filled one.

**Figure 4 entropy-25-01066-f004:**
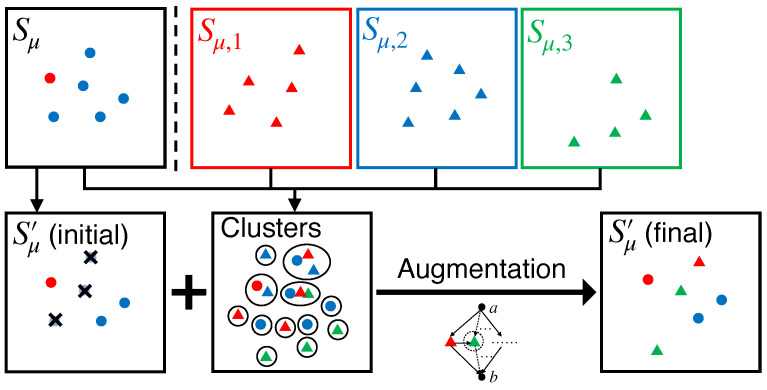
Illustration of post-processing in SFDM2. For each $\mu \in U^{\prime }$, an initial $S_{\mu }^{\prime }$ is first extracted from $S_{\mu }$ by removing the elements from over-filled groups. Then, the elements in all candidates are divided into clusters. The final $S_{\mu }^{\prime }$ is augmented from the initial solution by adding new elements from under-filled groups based on matroid intersection.

**Figure 5 entropy-25-01066-f005:**
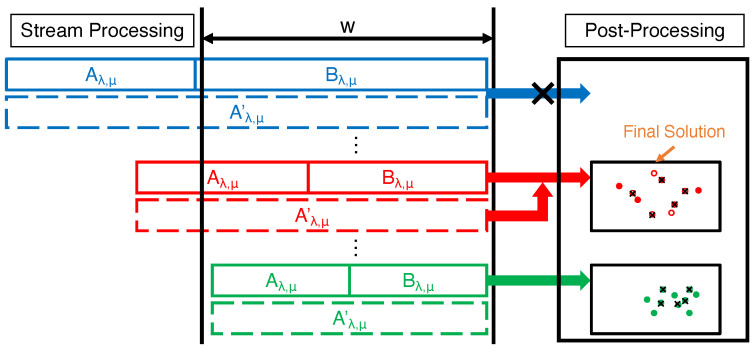
Illustration of the framework of sliding-window algorithms. During stream processing, two candidate solutions $A_{\lambda ,\mu }$ and $B_{\lambda ,\mu }$, along with their backups $A_{\lambda ,\mu }^{\prime }$ and $B_{\lambda ,\mu }^{\prime }$, are maintained for each guess λ,μ of OPT[W]. Then, during post-processing, the elements in $B_{\lambda ,\mu }$ and $A_{\lambda ,\mu }$ (or non-expired elements in $A_{\lambda ,\mu }^{\prime }$ if $A_{\lambda ,\mu }$ has expired) are passed to an existing algorithm for solution computation.

**Figure 6 entropy-25-01066-f006:**
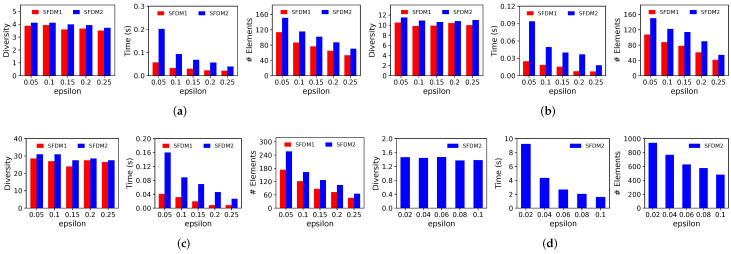
Performance of SFDM1 and SFDM2 with varying parameter ε on (**a**) Adult (Sex, m=2), (**b**) CelebA (Sex, m=2), (**c**) Census (Sex, m=2), and (**d**) Lyrics (Genre, m=15) when k=20.

**Figure 7 entropy-25-01066-f007:**
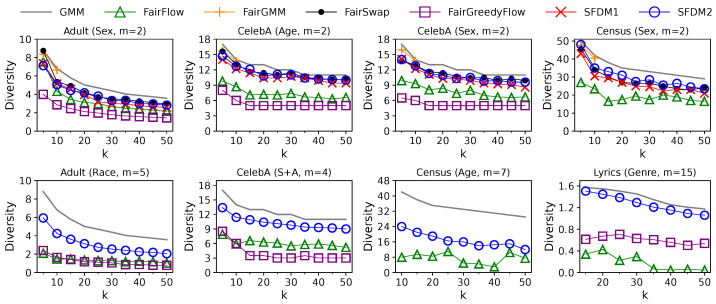
Solution quality of different algorithms in the streaming setting with varying solution sizes, *k*. The diversity values of GMM are plotted as gray lines to illustrate the “price of fairness”, i.e., the losses in diversity caused by incorporating the fairness constraints.

**Figure 8 entropy-25-01066-f008:**
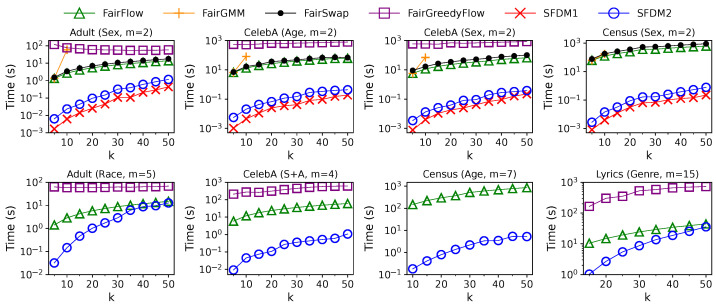
Update time of different algorithms in the streaming setting with varying solution sizes, *k*.

**Figure 9 entropy-25-01066-f009:**
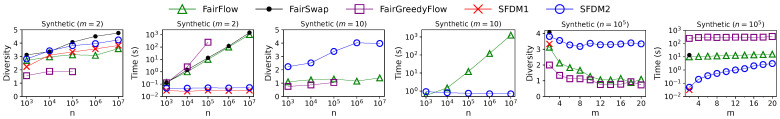
Solution quality and update time on synthetic datasets in the streaming setting with varying dataset sizes, *n*, and numbers of groups, *m* (k=20).

**Figure 10 entropy-25-01066-f010:**
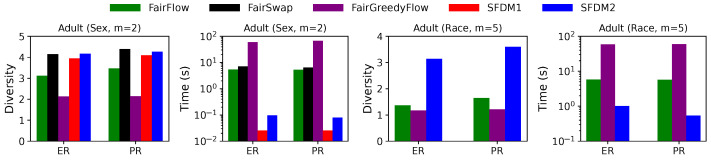
Comparison of different algorithms on *Adult* for equal representation (ER) and proportional representation (PR) when k=20.

**Figure 11 entropy-25-01066-f011:**
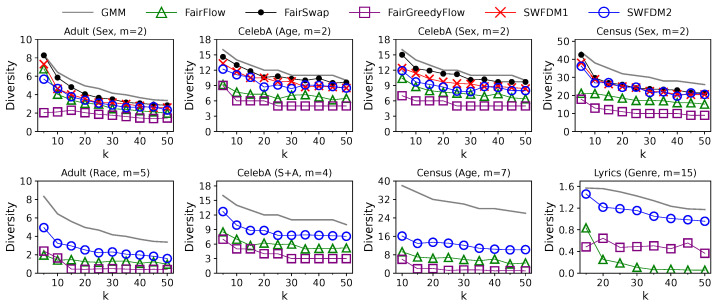
Solution quality of different algorithms in the sliding-window setting with varying solution size *k* (w=25 k for adult and 100 k for others). The diversity values of GMM are also plotted as gray lines to illustrate the “price of fairness”.

**Figure 12 entropy-25-01066-f012:**
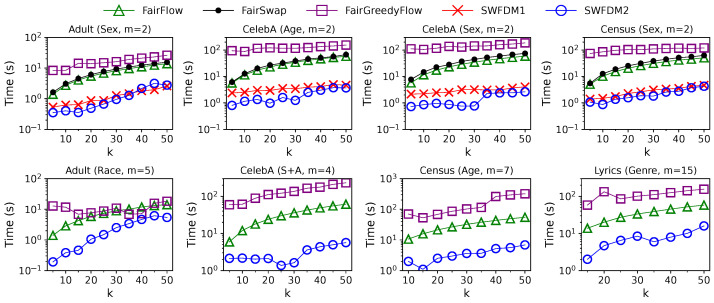
Update time of different algorithms in the sliding-window setting with varying solution size *k* (w=25 k for Adult and 100 k for others).

**Figure 13 entropy-25-01066-f013:**
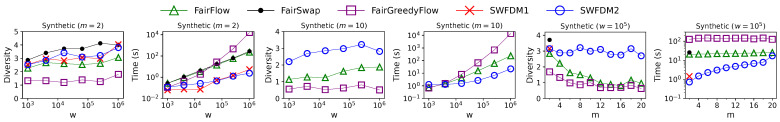
Solution quality and update time on synthetic datasets in the sliding-window setting with varying window size *w* and number of groups *m* (k=20).

**Table 1 entropy-25-01066-t001:** Statistics of datasets used in the experiments.

Dataset	Group	*n*	*m*	# Features	Distance Function
Adult	Sex	48,842	2	6	Euclidean
	Race		5		
	S + R		10		
CelebA	Sex	202,599	2	41	Manhattan
	Age		2		
	S + A		4		
Census	Sex	2,426,116	2	25	Manhattan
	Age		7		
	S + A		14		
Lyrics	Genre	122,448	15	50	Angular
Synthetic	-	$10^{3}$–$10^{7}$	2–20	2	Euclidean

**Table 2 entropy-25-01066-t002:** Overview of the performance of different algorithms in the streaming setting (k=20).

Dataset	Group	GMM	FairSwap		FairFlow		FairGreedyFlow		SFDM1			SFDM2		
		Diversity	Diversity	Time (s)	Diversity	Time (s)	Diversity	Time (s)	Diversity	Time (s)	#Elem	Diversity	Time (s)	#Elem
Adult	Sex	5.0226	4.1485	7.06	3.1190	5.45	2.1315	59.99	3.9427	0.0256	90.2	4.1710	0.0965	120.4
	Race		-	-	1.3702	5.82	1.1681	58.59	-	-	-	3.1373	1.0175	312.3
	S + R		-	-	1.0049	6.55	0.8490	60.92	-	-	-	2.9182	3.0914	620.6
CelebA	Sex	13.0	11.4	35.13	8.4	22.97	5.0	705.4	9.8	0.0188	87.2	10.9	0.0410	122.3
	Age		11.4	31.69	7.2	23.06	5.0	657.4	10.4	0.0225	94.6	10.8	0.0591	128.0
	S + A		-	-	6.3	24.17	3.5	312.8	-	-	-	10.4	0.1124	193.1
Census	Sex	35.0	27.0	372.3	17.5	254.6	-	-	27.0	0.0321	121.5	31.0	0.0931	163.0
	Age		-	-	8.5	294.0	-	-	-	-	-	21.0	0.8671	676.0
	S + A		-	-	5.0	347.5	-	-	-	-	-	19.0	3.7539	1276.0
Lyrics	Genre	1.5476	-	-	0.4244	14.84	0.6732	302.6	-	-	-	1.4463	2.6785	677.2

**Table 3 entropy-25-01066-t003:** Overview of the performance of different algorithms in the sliding-window setting (k=20; w=25 k for Adult and w=100 k for other datasets).

Dataset	Group	GMM	FairSwap		FairFlow		FairGreedyFlow		SWFDM1			SWFDM2		
		Diversity	Diversity	Time (s)	Diversity	Time (s)	Diversity	Time (s)	Diversity	Time (s)	#Elem	Diversity	Time (s)	#Elem
Adult	Sex	4.9598	4.0568	6.11	3.0660	5.47	2.0501	13.95	3.5445	0.870	431.5	3.4052	0.489	551.8
	Race		-	-	1.2364	5.87	0.4162	7.71	-	-	-	2.5212	1.056	616.1
	S + R		-	-	0.9105	6.43	0.3276	4.15	-	-	-	1.7843	1.027	799.6
CelebA	Sex	12.0	11.3	28.12	7.7	23.22	6.0	138.4	9.7	2.526	427.7	8.7	0.875	560.6
	Age		10.6	26.99	7.3	23.11	6.0	121.5	10.5	2.966	418.5	8.7	0.976	537.2
	S + A		-	-	6.2	24.10	4.0	112.1	-	-	-	8.8	2.119	864.1
Census	Sex	32.0	30.0	26.33	18.0	21.49	11.0	109.9	29.0	2.593	377.0	28.0	1.614	397.0
	Age		-	-	5.0	22.49	2.0	76.15	-	-	-	13.0	2.568	646.0
	S + A		-	-	2.0	24.11	5.0	128.4	-	-	-	13.0	3.077	796.0
Lyrics	Genre	1.5586	-	-	0.2522	20.12	0.6432	133.4	-	-	-	1.2166	4.694	1132.4

## Data Availability

Real-world datasets that we use in our experiments are publicly available. The code for generating synthetic data and for our experiments is available at https://github.com/yhwang1990/code-FDM (accessed on 12 July 2023).

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
