# Peer review of "Fair Max–Min Diversity Maximization in Streaming and Sliding-Window Models†"

_entropy, 2023, doi:10.3390/e25071066_

Round 1

Reviewer 1 Report

In this article, the authors propose some approximation algorithms for problems related to suitable streaming model. Furthermore, they introduce a sliding-window model algorithm, where only the latest elements in the stream are considered for computation to capture the recency of the data. 

This is an interesting article, which has some issues to address. first of all, the manuscript should be proofread to remove the typographical errors and imprecisions throughout the paper. The theoretical analysis is well designed and helpful. I would, however, suggest expanding the discussion/introduction of the theoretical analysis on page 11. Figure 6 should be further explained and justified.

There are some minor typos

Author Response

Dear Reviewer 1,

Thanks for your comments to help improve this manuscript.

We have revised the following parts of the manuscript according to your comments.

(1) We carefully proofread it to fix as many typos and imprecise expressions as possible.

(2) We expand the discussion of the theoretical analysis on Page 11.

(3) We add additional explanations and justifications for Figure 6.

Best Regards,

Yanhao, Francesco, Michael, and Jia

Reviewer 2 Report

This paper proposes two approximation algorithms for addressing the problem of fair max-min diversity maximization (i.e., maximizing the minimum distance for dissimilarity) in the streaming model and sliding-window model. They conduct four publicly available real-world datasets, and their experimental results show that the proposed algorithms provide solutions of comparable quality to the state-of-the-art offline algorithms while running in the streaming and sliding-window settings. From the study results, this paper has an interesting topic and good performance, but it has some minor remarks that should be firstly addressed in the revised manuscript, as follows:

1.      Although this paper has a logical and well-prepared structure with clear descriptions from technical perspective, I suggest that the authors should strengthen the description of the application level in order to enhance the paper readability for general readers.

2.      Moreover, significance of the research and motivation for this study is not clear.

3.      Please check the following statements:

(1)   In Page 2, the “w.r.t.” should not be presented in abbreviation in its first presentation. 

(2)   I suggest it is unnecessary to do the repeat abbreviation, such as fair max-min diversity maximization (FDM), please recheck them and all the cases should be revised.

(3)   For Pages 6, 8, 11, 12, and 16, they have additional character of square (â–¡), please correct them.

(4)   In end of Page 12, the “Section 4 and 5.1” should be revised to “Sections 4 and 5.1”.

(5)   In Page 13, the “Lines 2–3” and “Lines 7–8” should be changed to “Lines 2 and 3” and “Lines 7 and 8”.

(6)   In Page 22, the “Fig. 11–12” should be revised to “Figs. 11 and 12”.

Author Response

Dear Reviewer 2,

Thank you for your helpful comments on improving the manuscript.

We have revised the following parts of the manuscript according to your comments.

(1) We improve the description of the real-world applications of diversity maximization in the introduction, including its usage in data summarization, information retrieval, and feature selection.

(2) We highlight the significance of the research and the motivation for this study in the introduction from three aspects: (i) the importance of diversity maximization in real-world applications; (ii) fairness is an important issue in machine learning and data mining problems including diversity maximization; (iii) streaming and sliding window models are essential to scale the problem to massive data analysis.

(3) We have corrected all the typos and imprecise expressions as listed. As for additional characters of squares, they are the symbols to indicate "ends of proofs" and are added by the LaTex compilation system.

Best Regards,

Yanhao, Francesco, Michael, and Jia